# PHYSICS-CONSTRAINED GRAPH SYMBOLIC REGRESSION

## ABSTRACT

As data-driven scientific discovery increasingly demands explainable over 'black-box' machine learning (ML) methods, Symbolic Regression (SR) that derives analytical expressions can help identify key functional dependencies in complex systems. However, traditional SR methods often suffer from (a) inefficient exploration due to their inability to compress the search space of equivalent expressions, and (b) non-physical solutions that violate fundamental physics constraints. We here introduce a symmetric invariant representation of candidate analytical expressions using a Symbolic Graph (SG), on which the Symbolic Graph Neural Network (SGNN) encodes operators, symmetries, constraints and constant fitting knowledge. We further develop reinforcement learning (RL) algorithms with Monte-Carlo Tree Search (MCTS) on our SGNN for SR. Such a physics-constrained graph symbolic regression (PCGSR) method effectively compresses the search space for efficient SR. Experiments on synthetic and real-world scientific datasets demonstrate the efficiency and accuracy of our PCGSR in discovering underlying expressions and adhering to physical laws, yielding physically meaningful solutions.

## 1 INTRODUCTION

Symbolic regression (SR) (Angelis et al., 2023; Makke & Chawla, 2024) is an approach to unveil the inherent dependencies governing the system under study in a symbolic form. Unlike traditional regression techniques that adhere to predefined forms (e.g., linear, polynomial, exponential), SR operates without assuming any specific model form. Instead, SR explores the space of closed-form mathematical expressions, using variables and operations to find the most suitable analytical expressions that capture the relationships of the underlying observed data. This approach balances accuracy and interpretability, highlighting its potential in advancing AI for scientific discovery (Wang et al., 2019). Unlike SR, widely adopted black-box methods such as neural networks lack transparency, making it difficult to understand the underlying mechanisms and resulting in potentially perpetuating biases or inaccuracies in scientific research.

Symbolic regression (SR) has inspired extensive research due to its flexibility and expressive power. Early methods focus on Genetic Programming (GP) (Koza, 1994; Schmidt & Lipson, 2009; Fortin et al., 2012; Hernandez et al., 2019), which search the candidates by evolving expressions through selection, mutation, and crossover, avoiding brute-force methods like SISSO (Ouyang et al., 2018) or basis dependent methods like SINDy (Brunton et al., 2016). However, GP methods scale poorly, often yield overly complex solutions, and are sensitive to hyperparameters (Petersen et al., 2020). In contrast, modern SR methods leverage deep learning (DL) and reinforcement learning (RL) to enhance heuristic search efficiency. DSR (Petersen et al., 2020) uses a recurrent neural network (RNN) and risk-seeking policy gradients, excelling in simple tasks but struggling with complex ones due to limited exploration. NGGP (Mundhenk et al., 2021) combines RL with genetic programming, outperforming both GP and DSR on several benchmarks. MCTS-based methods (Świechowski et al., 2023; Sun et al., 2022; Kamienny et al., 2023) achieve a superior exploration-exploitation balance, delivering state-of-the-art performance. Deep generative models (Valipour et al., 2021; Biggio et al., 2021) excel at inference but lack adaptability for out-of-distribution datasets due to their static pretraining. This paper introduces a neural-guided MCTS framework, combining the pure Deep-RL's exploitation capability and pure MCTS's exploration capabilities, and further enhances

efficiency and applicability through state space reduction by capturing invariances and constraints, as detailed below.

**Expression Representations and Symmetries**

Despite extensive research addressing the challenge of exponentially increasing search space in SR with growing complexity (number of operators and variables involved in an expression), less focus has been given to the representation of expressions themselves. Most works (Petersen et al., 2020; Sun et al., 2022; Hernandez et al., 2019) use expression trees (Makke & Chawla, 2024) to convert expressions into input features for SR methods. However, such representations fail to capture the symmetries and invariances within expressions, leading to redundant states for equivalent expressions and reduced exploration efficiency, especially for sequential encoding methods like RNNs. AI-Feynman (Udrescu & Tegmark, 2020; Udrescu et al., 2020) attempted to address this by pre-training neural networks to capture modularities and symmetries to simplify SR problems into smaller sub-problems for brute-force search. However, this approach requires extra pre-training and the brute-force search remains inefficient for complex systems. To address these limitations, we propose a symbolic graph (SG) representation that not only inherently captures symmetries and invariances without pre-training, but also uniquely identifies operator directions (e.g., $-, \div, \wedge$) through edge features. This representation significantly compresses the search space, enhancing the exploration efficiency and accelerating the convergence.

**Symbolic Regression with Constraints**

Constraints exist in nearly all SR problems, especially for real-world problems involving constraints governed by the fundamental physical laws in natural science and other principles/rules in different fields. SR methods that fail to account for these constraints can yield meaningless results. Existing approaches (Udrescu & Tegmark, 2020; Tenachi et al., 2023; Keren et al., 2023) typically address this by either hand-crafted priors or penalty functions to incorporate domain-specific knowledge. Though domain-specific priors can prevent invalid expressions from generating, domain-specific penalty functions as well as other hidden constraints can lead to sparse rewards, making it challenging for SR methods to converge. To overcome this issue, here we propose to employ a symbolic graph neural network (SGNN) on SG, trained by MCTS to provide the encoding of prior knowledge for domain-specific penalty functions and other hidden constraints, guiding MCTS simulations to reduce reward sparsity. This approach ensures that the generated expressions adhere to the necessary constraints while also mitigating overfitting, thereby enhancing the practical applicability and robustness of SR in real-world scenarios.

In summary, we present Physics-Constrained Graph Symbolic Regression (PCGSR), a novel SR methodology designed to address the challenges of inefficient exploration caused by redundant representations of equivalent expressions and the limitations of random policy in MCTS simulation. Our approach incorporates physics constraints directly into the search process to produce physically meaningful results. We achieve these through 1) SG representations utilized in MCTS and SGNN to effectively compress the search space by capturing symmetries and invariances within equivalent expressions; 2) SGNN encoding that embeds inductive biases from MCTS and physics constraints into the SGNN representation; 3) SGNN-guided MCTS that replaces the random policy in MCTS simulation with an SGNN-based policy, enabling efficient exploration with encoded inductive biases and physics constraints. We validate the effectiveness of PCGSR through benchmarking on widely recognized synthetic datasets and a real-world application in materials science. The results highlight the practical utility and robustness of our approach, demonstrating its capability to tackle complex problems in real-world scientific discovery.

## 2 PROBLEM STATEMENT

Given a dataset $\mathcal{D} = \{(\boldsymbol{x}_1, y_1), (\boldsymbol{x}_2, y_2), \ldots, (\boldsymbol{x}_n, y_n)\}$, SR aims to find a mathematical expression $f$ to map the input feature vector $\boldsymbol{x}_i$ to the corresponding output value $y_i$ for each sample with the minimum error over all data points in $\mathcal{D}$:

$$\min_f \sum_{i=1}^{n} L(f(\boldsymbol{x}_i), y_i), \tag{1}$$

where $L(\cdot, \cdot)$ is a loss function that measures the difference between the predicted and actual output values as the error. $f$ belongs to a space of mathematical expressions constructed using a predefined dictionary set $\mathcal{Q}$ of $n$ mathematical operators $\phi$. That is, $\mathcal{Q} = \{\phi_0, \phi_1, \ldots, \phi_n\}$ and $f = [\phi_i | \phi_i \in \mathcal{Q}]$. In this paper, every input feature $\boldsymbol{x}_i$ is regarded as an operator. We also include operator $const$ for inserting constants and $trans$ for transplantation strategy by default, detailed in Appendix B.1.

## 3 METHODS

The main challenge of symbolic regression (SR) originates from its infinite search space for mathematical expressions, as it encompasses extensive combinations of operators and features to compose expressions. To search for such a space efficiently, we propose our Physics-constrained Graph Symbolic Regression (PCGSR) methodology, which strategically leverages symmetries, invariances, and constraints to construct a condensed search space for accurate and physically meaningful governing expressions. It is achieved through the following three key innovations :

- SG representations for expressions to capture symmetries and invariances within them;
- SGNN that encodes physics constraints, hidden constraints, and constraints fitting knowledge;
- MCTS with SGNN-integrated simulation under constraints, that yields less sparse rewards, light cost for constants fitting, and physically meaningful solutions to real-world problems.

### 3.1 SYMBOLIC GRAPH REPRESENTATION

Sampling expressions in SR can be modeled as a Markov Decision Process (MDP), where a new operator or feature is iteratively sampled based on the current expression state. The common approach for representing this state is the expression tree (ET) (Makke & Chawla, 2024), as shown in Figure 1. In this structure, operators are modeled as nodes and edges represent relationships between operators and their operands, with the tree growing from outer functions to inner ones. This approach reflects the sequential construction of expressions through iterative sampling steps.

However, the ET representation, widely used in many existing SR methods (Petersen et al., 2020; Mundhenk et al., 2021; Sun et al., 2022; Hernandez et al., 2019), has a significant limitation—it fails to account for invariances regarding the operator generation sequence of expressions. For example, as shown in Figure 1, two symmetric expressions (mathematically equivalent with the same nodes) can have different operator-generating orderings, leading to distinct sibling relationships and different tree structures. This lack of invariance modeling results in redundant representations for equivalent expressions (common in polynomials or products), which reduces the learning efficiency of SR models. This issue is particularly problematic for RL approaches, which may struggle to converge when faced with an unnecessarily large and diverse state space.

To overcome these challenges, we propose a novel expression representation called the Symbolic Graph (SG), denoted as $G$, which converts the ET into an undirected graph-based representation (Figure 2). In this model, we retain the same node structure for unary operators, while binary operators are represented through a combination of node and edge features, effectively unifying operators like "$+$" and "$-$" or "$\times$" and "$\div$". Additionally, we differentiate the operands in directed binary operations such as "$-$", "$\div$", and "$\%$". This approach inherently captures symmetries and invariances, reducing both the search space for SR and the state space for RL. The benefits include: 1) expanding commutative invariance for polynomial and product terms, 2) preserving permutation invariance in the generation order by grouping consecutive operations at the same level, and 3) enhancing the representation precision by uniquely distinguishing operands in directed operators. Appendix B.2 provides a detailed description of our SG representation.

### 3.2 SYMBOLIC GRAPH NEURAL NETWORK

A key innovation of our PCGSR lies in the introduction of an efficient exploration strategy for the condensed search space generated by the SG representation. To enable this, we propose the Symbolic Graph Neural Network (SGNN), based on Graph Convolutional Networks (GCN) (Kipf & Welling, 2016; Xie & Grossman, 2018), which enhances the search efficiency by incorporating inductive bias to replace the random rollout of MCTS's simulation. GCN, a prominent architecture

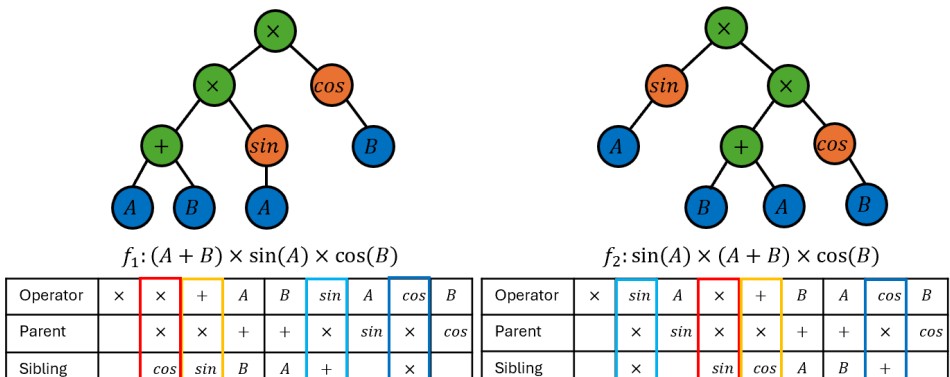

$f_1: (A+B) \times \sin(A) \times \cos(B)$      $f_2: \sin(A) \times (A+B) \times \cos(B)$

| Operator | × | × | + | A | B | sin | A | cos | B |
|---|---|---|---|---|---|---|---|---|---|
| Parent | | × | × | + | + | × | sin | × | cos |
| Sibling | | cos | sin | B | A | + | | × | |

| Operator | × | sin | A | × | + | B | A | cos | B |
|---|---|---|---|---|---|---|---|---|---|
| Parent | | × | sin | × | × | + | + | × | cos |
| Sibling | | × | | sin | cos | A | B | + | |

Figure 1: An example of not capturing symmetries in sequential-encoding expression trees. The left expression $f_1$ and the right expression $f_2$ are two symmetric expressions but generated in a different order as in the given tables. Any sequential encoding based on the semantic text tokens or the tree structures will yield different representations. The colored box highlights different sibling relationships in the tree structure which breaks the permutation invariance in generating the expression.

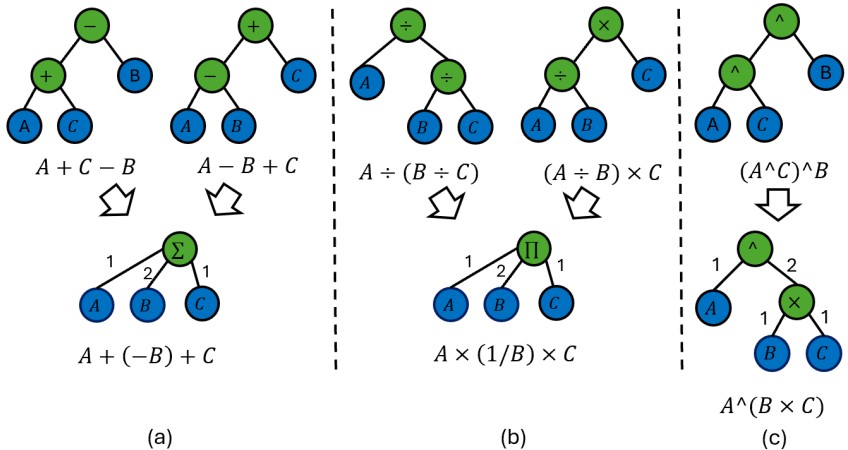

(a)        (b)        (c)

Figure 2: An example of SG encoding symmetries and invariances by: (a) replacing consecutive addition and subtraction into a 'sum' $\Sigma$ operator, where the edge feature "1" represents an added term and "2" represents a subtracted term; (b) replacing consecutive multiplication and division operators into a 'product' $\Pi$ operator, where the edge feature "1" represents a multiplied term, and "2" represents a divided term; and (c) simplifying consecutive exponentials by combining them into a single exponential, with the power node representing the multiplication of powers, where the edge feature "1" represents the base and "2" represents the exponent.

for processing graph-structured data, serves as the foundation of SGNN, as detailed in Appendix B.3. Using the SG representation $G$ for an expression, the node set $\mathcal{V} = \phi_i$ represents the operators, and the edge set $\mathcal{E} = e_{ij}$ captures relationships between operators and operands. SGNN encoding of $G$ is formally expressed as:

$$\text{SGNN}(G\{\mathcal{V}, \mathcal{E}\}) = (\pi_\theta|_{\mathbf{P}}(G), \pi_\theta|_r(G)) = (\mathbf{P}, r) \tag{2}$$

where the output $\mathbf{P}$ is a prior probability matrix, with each row representing the prior probability distribution for each operator $\phi$ to be added at a given node during MCTS simulations. The output $r$ is the predicted reward value. $\pi_\theta|_{\mathbf{P}}$ and $\pi_\theta|_r$ are models predicting $\mathbf{P}$ and $r$ given $G$ with trainable parameters $\theta$ of SGNN. SGNN is trained by self-learning with MCTS, incorporating physics constraints and inductive bias to guide MCTS simulations effectively. This approach significantly reduces reward sparsity due to constraints and boosts the overall search efficiency.

### 3.3 SGNN-GUIDED MONTE-CARLO TREE SEARCH

Attributed to SR's expansive search space and MDP property, Monte-Carlo Tree Search (MCTS) has emerged as one of the promising methods for SR (Sun et al., 2022; Kamienny et al., 2023). Offering efficient sampling and a robust exploration-exploitation trade-off, MCTS exhibits resilience against local optimal traps, a challenge faced by policy gradient methods. To leverage the capabilities of our SG representation and SGNN encoding, we employ SGNN-guided MCTS methods inspired by the approach taken in AlphaGo Zero (Silver et al., 2017; Nair, 2017).

We model the expression generation process as a finite-horizon sampling trajectory $\tau = \{s_0, a_0, s_1, a_1, \ldots, s_t, a_t, \ldots, a_{H-1}, s_H\}$, with a maximum complexity of $H$. At step $t$ we define the state $s_t$ to be the current SG representation $G_t$ with the sampling node. Action $a_t$ is the newly added operator or input feature $\phi_t$. When a trajectory $\tau$ is complete (attain the maximum complexity $H$ or finish the expression in closed form), we obtain the finalized expression as a function $f_\tau$. Then we can evaluate $\tau$ through the reward $R(\tau) = 1/(1 + \text{NMAE})$, where NMAE is the normalized mean absolute error defined as:

$$\text{NMAE} = \frac{1}{\sigma_y} \frac{1}{n} \sum_{i=1}^{n} |y_i - f_\tau(X_i)| \tag{3}$$

To sample $\tau$, at each step $t$ in $\tau$, we do one batch of MCTS for each state $s_t$ to update MCTS policy $\pi_M$ so that we can sample $a_t \sim \pi_M(s_t)$. During each batch, an MCTS will simulate a trajectory $\tau_t = \{s_t^0, a_t^0, s_{t+1}^1, a_{t+1}^1, \ldots, s_{t+i}^i, a_{t+i}^i, \ldots, a_{H-1}^{H-t-1}, s_H^{H-t}\}$ from the root state $s_t$, where the superscript denotes the step of the MCTS simulation and subscript denotes the current complexity. We will record $Q(s, a)$ (the expected action value for taking action $a$ from $s$), $N(s, a)$ (the number of times taking action $a$ from state $s$ across simulations), $\mathbf{P}(s, \cdot)$ (prior probability distribution of taking action from state $s$), and $R(s)$ (the expected reward of state $s$), for $(s, a)$ pair that MCTS has traversed through during this simulation. Specifically, MCTS will do the following four steps to simulate $\tau_t$:

**Selection:** During this step, MCTS will start from the root state $s_t$ and select the next step iteratively before arriving at an expandable node or terminal node, with the maximum Upper Confidence Bounds (UCB) policy $argmax_a\text{UCB}(s, a)$. Define $b$ to be the next possible action and $c_{puct}$ to be a hyperparameter controlling the exploration rate, we have $\text{UCB}(s, a)$ as:

$$\text{UCB}(s, a) = Q(s, a) + c_{puct}P(s, a)\frac{\sqrt{\sum_b N(s, b)}}{1 + N(s, a)}. \tag{4}$$

**Expansion:** If MCTS traverses to a visited node with unvisited children, we call it an expandable node. We will select unvisited children according to UCB.

**Simulation:** This is the part where SGNN will guide UCB. When traversing to an unvisited node, we will use SGNN encoding with a controlling coefficient $\epsilon$ (explained in Appendix B.4). That is, with the possibility $\epsilon$, we will have the uniform prior and calculate $R(s)$ through random rollout as the naive MCTS does. Otherwise, we obtain $(\mathbf{P}(s, \cdot), R(s)) = (\pi_\theta|_{\mathbf{P}}(s), \pi_\theta|_r(s))$ through SGNN encoding according to Equation 2 instead.

**Backpropagation:** After we obtain the R(s) of the unvisited node, we will increase $N(s, a)$ by one and update $Q(s, a) = (1/N(s, a)) \sum_{s'|s,a} R(s')$, for those $(s, a)$ pairs that have been traversed through. $s'$ represents the next state of $s$ by taking action $a$.

After we finish one batch of MCTS, we can update $\pi_M = N(s_t, \cdot)/\sum_b N(s_t, b)$ to step next in the trajectory. Once we complete a trajectory $\tau$, we subsequently update the parameters $\theta$ of SGNN by minimizing the loss function:

$$l = \sum_t ((R(\tau) - \pi_\theta|_r(s_t))^2 - \pi_M(s_t) \log \pi_\theta|_{\mathbf{P}}(s_t)). \tag{5}$$

Appendix B.5 outlines the proposed SGNN-guided MCTS algorithm. Notably, within this framework, SGNN encodes constraints and constants fitting knowledge through $(\mathbf{P}(s, \cdot), R(s)) = (\pi_\theta|_{\mathbf{P}}(s), \pi_\theta|_r(s))$, guiding the MCTS simulation. The SR representation also captures invariances, ensuring that equivalent expressions share the same state $s$ for MCTS (e.g. $Q(s, a)$ and $N(s, a)$) and SGNN (e,g. $\mathbf{P}(s, \cdot), R(s)$). Without this mechanism, MCTS and SGNN would treat equivalent expressions differently, resulting in reduced optimization efficiency due to divergent values for equivalent states.

## 3.4 Constraints Incorporation

Constraints serve as crucial prior knowledge for SR. Their significance lies not only in confining exploration to valid areas but also in guiding SR to uncover meaningful and robust solutions for real-world applications. In our PCGSR model, we classify constraints into two categories:

- **Pre-constraints:** Simple constraints testable during operation sampling, such as: *basic mathematical rules*, *void operations*, *maximum complexity* (the number of nodes in SG), and *hand-crafted prior* for *a priori* known physics constraints.
    - **Strategy:** Pre-constraints are incorporated by zeroing out probabilities of actions violating constraints, preventing invalid expressions during operation sampling.
- **Post-constraints:** Complex constraints requiring a complete expression for evaluation, such as *hidden constraints* from SR problems (e.g. $\log(A+?)$ will be invalid in the real-number realm if "?" is further sampled to be "$f(B)$" and $A + f(B) < 0$), and hand-crafted penalty functions for *a priori* known physics constraints (e.g., $f(r \to 0) = \infty$ in Section 5.2).
    - **Strategy:** Post-constraints are incorporated by penalizing invalid outputs with zero rewards after generating a complete expression.

PCGSR's flexible constraint incorporation strategies allow users to define and categorize custom constraints for effective integration. Traditional SR methods cannot seamlessly incorporate post-constraints as pre-constraints to prevent invalid expressions during sampling, as these require complete expressions for evaluation. This limitation leads to sparse rewards in search spaces with complex constraints (e.g., Section 5), posing a significant challenge for sampling-based methods like naive Monte-Carlo Tree Search (MCTS) or Genetic Programming (GP). These methods struggle to adapt to constraint violations during certain training phases (e.g., random simulation in MCTS or random crossover and mutation in GP). PCGSR overcomes these challenges by enabling self-learning within MCTS through Equation 5, allowing physics and hidden post-constraint insights to be effectively incorporated into pre-constraint strategies during the operation sampling phase of MCTS. By leveraging the predicted prior **P** from Equation 2, PCGSR mitigates reward sparsity, enhances search efficiency, and proves particularly effective for real-world problems involving intricate physics constraints, significantly boosting its practical applicability.

## 3.5 Cost Effectiveness

PCGSR excels in balancing exploration and exploitation through its neural-guided MCTS framework, outperforming the pure Deep-RL-based approach DSR (Petersen et al., 2020) and the pure sampling-based method SPL (Sun et al., 2022), as shown in Section 4. It is also significantly more cost-effective than DSR, SPL, and the neural-guided GP method NGGP (Mundhenk et al., 2021), primarily due to its efficient handling of the most computationally expensive aspect of SR—constants fitting (detailed in Appendix B.1). Unlike SPL and NGGP, which rely on costly constants fitting for reward evaluation at every step, PCGSR leverages SGNN's lightweight forward and backward passes to encode and predict rewards $r$ in Equation 2. Additionally, SG representations reduce the search space by capturing symmetries and invariances, thereby minimizing the number of constants fitting steps required. Finally, PCGSR's computational cost is dynamically adjustable via the controlling coefficient $\epsilon$, as outlined in Appendix B.4, ensuring adaptability to varying problem complexities.

## 4 Experiments

### 4.1 Synthetic Dataset Benchmarking

We have evaluated PCGSR on diverse benchmark datasets, including 1) the Feynman dataset (Udrescu & Tegmark, 2020) in SRBench (La Cava et al., 2021), 2) Nguyen's SR benchmark dataset Uy et al. (2011), and 3) Nguyen's SR benchmark with constants dataset (Petersen et al., 2020). We compare the results by PCGSR and state-of-the-art (SOTA) baselines in this section (Feynman) and Appendix C.2 (Nguyen's). In these benchmarks, we consider the following baseline models: Symbolic physics Learner (SPL) (Sun et al., 2022), an SR model based on naive MCTS method; Neural-Guided Genetic Programming (NGGP) (Mundhenk et al., 2021), an SR

model based on RNN-based policy gradients with GP tuning; Deep symbolic regression (DSR) (Petersen et al., 2020), an SR model based on RNN-based risk-seeking policy gradients, AI Feynman 2.0 (Udrescu et al., 2020), brute-force method with a pre-trained neural network to capture symmetries and modularity; and the traditional GP method with `gplearn` (Stephens, 2016). We summarize the introduction of benchmarking datasets and our experimental settings in Appendix C.1.

| Model | PCGSR | SPL | NGGP | DSR | AI Feynman 2.0 | GP |
|---|---|---|---|---|---|---|
| **Recovery Rate** (%) | **62.18 ± 3.00** | 58.93 ± 3.73 | 60.22 ± 2.27 | 23.62 ± 2.28 | 51.26 ± 5.82 | 20.17 ± 3.21 |
| **Complexity** | **30.56** | 32.48 | 36.57 | 22.78 | 42.01 | 46.05 |

Table 1: Performance comparison of PCGSR with baseline methods on the Feynman dataset: We report the average recovery rate with the 95% confidence interval as well as the average expression complexity. The recovery rate is the ratio of ground-truth equivalent solutions in mathematics to the total equations in the dataset.

| Model | PCGSR | MCTS-SG | MCTS-GNN | MCTS |
|---|---|---|---|---|
| SG | ✓ | ✓ | × | × |
| SGNN | ✓ | × | ✓ | × |
| **Number of Evaluations** | **96,221** | 121,853 | 237,923 | 285,284 |
| **Training time** ($s$) | **1847.4** | 2485.8 | 4306.4 | 5848.3 |

Table 2: The average recovery rate, the average number of evaluations, and the average training time for the highest recovery rate for ablation studies on eight Feynman equations with two features. The recovery rate is the ratio of ground-truth equivalent solutions in mathematics to the total of parallel experiments for the same equation.

In Table 1 for benchmarking on the Feynman dataset, PCGSR presents the best expressive power (average recovery rate of 62.18%, average training time of 4.3 hours) with the lowest average complexity, outperforming the average recovery rate and average training time of hybrid RL with GP method NGGP (60.22%, 5.7 hours), the naive MCTS-based SPL (58.93%, 5.2 hours), the pure RNN-based RL method DSR (23.62%, 6.1 hours), the previous SOTA methods in SRBench, AI Feynman 2.0 (51.26%, 7.7 hours) and the traditional GP method (20.17%, 3.8 hours). We have also benchmarked these baselines in the Nguyen's SR benchmark dataset and Nguyen's SR benchmark with constants dataset in Appendix C.2, which further verifies the SOTA performances by PCGSR. These results demonstrate the potential of PCGSR in capturing the underlying dependency relationships of complex systems with analytical solution expressions.

## 4.2 ABLATION STUDY

Furthermore, we perform ablation studies of PCGSR with or without SG representation and SGNN encoding, to assess the efficacy of constituting components in PSGSR by Table 2. In this study, we choose eight equations involving two features from the Feynman dataset according to Appendix C.3, which are under the same experimental settings and are conducted 10 times on each equation. Besides the proposed PCGSR, we include the following adapted methods for ablation studies: 1) MCTS-SG, adapted from PCGSR using random policy for MCTS's simulation rollout instead of SGNN encoding; 2) MCTS-GNN, adapted from PCGSR using expression tree representations instead of SG representations for MCTS and SGNN; 3) MCTS, adapted from PCGSR by removing both SG representations and SGNN encoding as a naive MCTS.

**Performance Analysis** Results in Table 2 show significant improvement with respect to both performance and search size (number of evaluations) when comparing SG-based PCGSR and MCTS-SG to non-SG-based MCTS-GNN and MCTS, proving the importance of capturing symmetries and invariances in expression representations for SR problems. On the other hand, SGNN encoding seems to only help reduce the search size but not obviously in performance when comparing SGNN-based PCGSR and MCTS-GNN to non-SGNN-based MCTS-SG and MCTS. This may be explained by not-so-sparse rewards in the explorations in this synthetic dataset. When incorporating complicated physics constraints in real-world applications, as in Table 3, PCGSR provides a significant improvement over non-SGNN-based MCTS.

**Efficiency Analysis** Table 2 highlights a significant search space reduction of 66.27% with PCGSR compared to standard MCTS, achieved through two key components. The invariance encoding in SG reduces the search space by 59.56% (PCGSR vs. MCTS-GNN) and 57.29% (MCTS-SG vs. MCTS), while SGNN further decreases the search size by 21.03% (PCGSR vs. MCTS-SG) and 16.60% (MCTS-GNN vs. MCTS). In terms of time efficiency (evaluations/second), the results are as follows: PCGSR (52.08), MCTS-SG (49.02), MCTS-GNN (55.25), and MCTS (48.78). MCTS-GNN achieves the highest efficiency due to its longer training phase and higher reliance on SGNN-guided simulations, as detailed in Appendix B.4. These findings demonstrate that SGNN enhances efficiency by reducing evaluation demands and avoiding costly coefficient fitting during simulations.

## 5 APPLICATIONS

In this section, we employ PCGSR in a real-world application for materials science, to show its efficacy in yielding physically meaningful solutions under constraints. Specifically, we focus on interatomic potential energy prediction for atomistic simulations, which is an important area for novel materials design and discovery. Though traditional methods for this area such as density functional theory (DFT) (Hohenberg & Kohn, 1964; Kohn & Sham, 1965), one of the first-principles methods, represent a critical and powerful solution, their ability is limited by the substantial computational cost and extensive memory requirements. As a consequence, surrogate ML models are under active development to expedite these simulations, among which SR has shown great promise for achieving both efficiency and interpretability. Appendix D.1 depicts the detailed problem backgrounds for this application, and Appendix D.2 depicts the practical usage of our PCGSR's solution.

### 5.1 PROBLEM SETTINGS

We adopt a dataset from first-principles molecular dynamics simulation of 32 copper atoms from Hernandez et al. (2019), which is detailed described in Appendix D.3. Our objective is to discover the interatomic potential energy function $f(\cdot)$ that effectively maps the atoms' pairwise distances $r$ to a total formation energy $E$. In this study, we consider four methods to learn the interatomic potential energy function: 1) a GNN-based black-box method CGCNN from Xie & Grossman (2018); 2) a genetic programming-based SR method from Hernandez et al. (2019); 3) our PCGSR in this paper; 4) and the MCTS method which is adapted from PCGSR but uses random policy instead of SGNN to guide MCTS's simulation, as an ablation study to assess the impact of SGNN in encoding physics constraints. Table 3 presents our comparison results for the formation energy prediction, showcasing the performance of our PCGSR alongside other models. Specifically, GP1 and GP2 are two fitted expressions by genetic programming reported in Hernandez et al. (2019). We summarize our experimental settings for baseline models in Appendix D.3.

### 5.2 PHYSICS CONSTRAINTS

To ensure the resulting $f(\cdot)$ is physically meaningful, we consider the following additional physics constraints besides the constraints mentioned in Section 3.4:

- **(1) Scalar Output** (*pre-constraint*): The output of $f(r)$ has to be a single scalar to match $E$. Because the input feature $r$ is a vector of variable length to different samples, a $\sum$ operator must be included in the expression before any $r$.
- **(2) Unit Consistency**(*pre-constraint*): A scalar *const* should be multiplied before any polynomial terms and in any power number in the exponential terms. It is defined to ensure that the unit calculation aligns with physical meaning, as *const* can introduce an extra unit as a coefficient. An example is given in Appendix D.5.
- **(3) Electrostatic Repulsion**(*post-constraint*): The energy approaches to infinite at an infinitesimal distance (i.e. $f(r \rightarrow 0) = \infty$). It is due to the Coulomb's law and a subtle consequence of the Pauli's exclusion principle of quantum mechanics. An explanation is given in Appendix D.5.

Note that CGCNN only satisfies the scalar output (1) constraint; GP1 and GP2 satisfy both the scalar output (1) and unit consistency (2) constraints; and MCTS and PCGSR satisfy all the constraints.

| Model | Comp | Expression $f(r)$ (eV/atom) | Cst | Train / Test | Transfer |
|-------|------|------|------|------|------|
| | | | | **MAE** (meV/atom) | |
| CGCNN | - | - | (2)× (3)× | **2.08** / 3.09 | 44.89 |
| GP1 | 21 | $\sum (r^{10.21-5.47r} - 0.21^r)s(r)$ $+0.97(\sum 0.33^r s(r))^{-1}$ | (2)✓ (3)× | 3.68 / 3.53 | 43.32 |
| GP2 | 28 | $7.33\sum r^{3.98-3.94r}s(r) + (27.32 - \sum$ $(11.13 + 0.03r^{11.74-2.93r})s(r))(\sum s(r))^{-1}$ | (2)✓ (3)× | 2.57 / 2.70 | 41.63 |
| MCTS-SG | 31 | $\sum (13.70r^3 + 27.18r^2 - 13.70re^r)e^{-0.98r^2}$ $+\sum 2.98r^{-1}e^{-12.87r}$ | (2)✓ (3)✓ | 3.91 / 3.65 | 36.31 |
| PCGSR | 24 | $\sum 6.04 \times 10^{-11}(r^3e^{3r} - r^{12})$ $+\sum 121.41(r^{-7} + 3r^{-8} - r^{-6})$ | (2)✓ (3)✓ | 2.41 / **2.63** | **34.15** |

Table 3: Analytical interatomic potential energy functions $f(r)$ fitted to DFT total energy using GP1, GP2, MCTS-SG, and PCGSR compared with the CGCNN model. Here, $f(r)$ is in the unit of $eV/atom$, and $s(r)$ is the smoothing function introduced in Eq (7) of Hernandez et al. (2019). "Comp" denotes the complexity of an expression, computed by the sum of nodes in the symbolic graph or the expression tree. "Cst" denotes the satisfied constraints defined in Section 5.2. "Train/Test" represents the training and testing MAE of the copper dataset. "Transfer" represents the transfer MAE of expressions directly tested on the newly generated dataset. Typically, DFT achieves MAE of 10–20 meV/atom for energy predictions

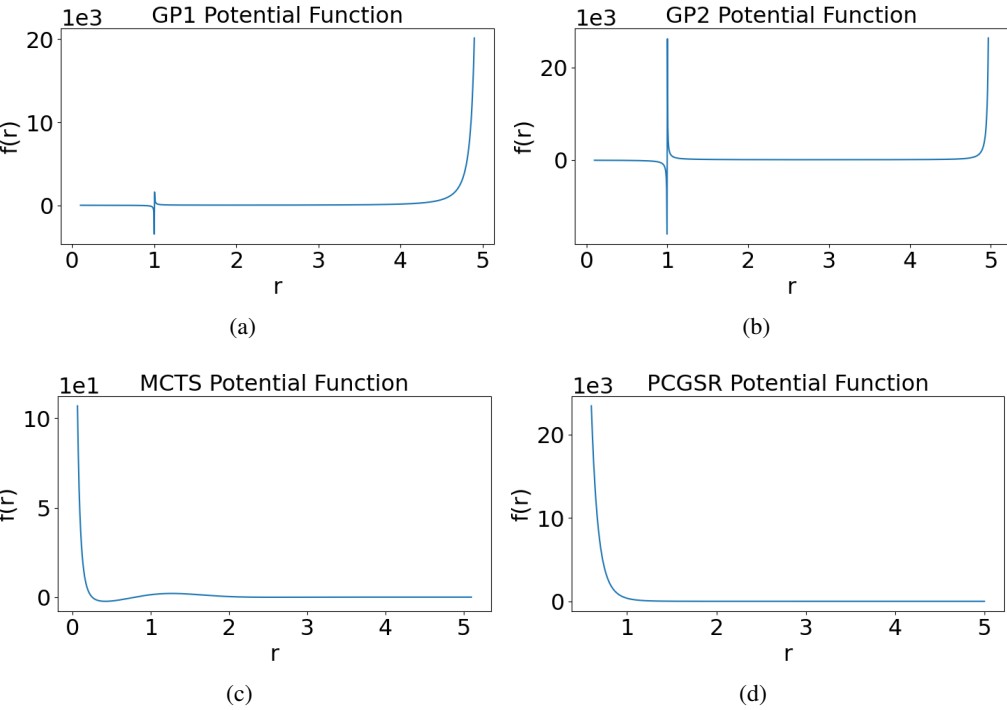

(a)

(b)

(c)

(d)

Figure 3: Interatomic potential energy functions $f(r)$ learned by (a) GP1, (b) GP2, (c) MCTS-SG, and (d) PCGSR, as a function of interatomic distance $r$ ranging between 0 to 5Å. The $x$-axis is the interatomic distance $r$ and the $y$-axis is the interatomic potential energy. Only MCTS-SG and PCGSR produce physically meaningful results, adhering to Pauli's exclusion principle, while GP1 and GP2 fail. A finite-valued model at short interatomic distances like GP1 and GP2 allows atoms to overlap, potentially causing the simulation to crash and leading to meaningless results violating physics laws.

### 5.3 Performances and Transferbility

In Table 3, the Mean Absolute Error (MAE) is computed based on linear regression results of the model expressions regarding the formation energy $E$, divided by the number of atoms (32) in the crystal. Figure 3 depicts the curves of the interatomic potential energy functions $f(r)$ according to the learned analytical expressions in Table 3 to illustrate the electrostatic repulsion that the energy goes to infinite $f(r) \to \infty$ at short interatomic distance ($r \to 0$). More details are described in Appendix D.6. Comparing training and testing MAEs for the copper dataset, PCGSR achieves the best testing MAE while preserving all physical constraints, where the $1/r^m$ terms reflect electrostatic repulsion at an infinitesimal distance. MCTS-SG, though satisfying all the constraints as PCGSR, its largest MAE and highest complexity highlight the impact of sparse rewards to naive MCTS during random rollout simulations. In contrast, PCGSR integrates SGNN which guides MCTS with encoded physics constraints that improves the likelihood of generating valid expressions during simulations. Although CGCNN has the lowest training MAE, its larger testing MAE indicates the potential overfitting risk, showing less robustness than SR methods such as PCGSR and GP2 in data-limited scenarios. GP1's low complexity comes at the cost of significantly poor performance, while GP2, despite the comparable predictive power to PCGSR, has higher complexity that makes the underlying dependencies less straightforward. Besides, none of CGCNN, GP1, or GP2 adheres to electrostatic repulsion constraint, resulting in less physically meaningful solutions at short interatomic distance. Overall, PCGSR demonstrates strong expressive power with lower complexity than GP methods and reduced overfitting compared to neural networks, while consistently yielding physically meaningful results. Plots for Table 3 results are included in Appendix D.4.

To further assess the transferability of the interatomic potential energy model, we create a new dataset of 100 samples by performing DFT-based molecular dynamics simulations where a volumetric compression of approximately 50% is applied to the unit cell of the original copper dataset. The new dataset thus consists of samples with shorter interatomic distances (bond lengths) between neighboring atoms, as described in Appendix D.7. This allows us to assess the transferability of our physics-constrained analytical model compared to other conventional machine learning models. The transfer MAE in Table 3 is obtained through zero-shot learning of the solution expressions directly tested on the newly generated dataset, where PCGSR and MCTS-SG show significantly lower MAEs. As electrostatic repulsion has a direct impact at shorter interatomic distances, PCGSR and MCTS-SG that integrate the physical constraints provide better generalization to shorter bond lengths on the new dataset. These results again verify that the incorporated physics constraints can prevent overfitting and foster the discovery of generalizable knowledge.

## 6 Conclusions and Future work

In this study, we identified two critical challenges in current Symbolic Regression (SR) models: the redundant representations for equivalent expressions due to an inability to capture inherent invariances, and the lack of a mechanism to incorporate post constraints, which increases reward sparsity and hinders the practical applications to real-world problems. To address these issues, we introduced the Physics-Constrained Graph Symbolic Regression (PCGSR) method, built upon the Monte-Carlo Tree Search (MCTS) framework for its balanced exploration-exploitation trade-off, and enhanced by the Symbolic Graph (SG) representation and Symbolic Graph Neural Network (SGNN). The SG representation effectively captures invariances, compressing the search space for SR, while SGNN encodes post constraints and constants fitting knowledge to guide MCTS simulations, thereby reducing reward sparsity and improving efficiency. Benchmark results on synthetic datasets and ablation studies demonstrate that the proposed SG representation significantly improves performance with the compressed search space. A real-world application in materials science, involving domain-specific physics constraints, further underscores the importance of SGNN in encoding these constraints to prevent overfitting and produce physically meaningful solutions.

PCGSR provides a comprehensive framework that enhances model accuracy, complexity management, robustness, and applicability in real-world problems. Looking ahead, expanding the scope of invariances to exploit deeper similarities between expressions could further simplify problem-solving within the current framework. Additionally, exploring the use of pre-trained SGNN models to encode general SR constraints for transfer learning could accelerate training and improve initialization, representing an intriguing avenue for future research.

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

APPENDIX

## A  COMPUTING RESOURCES

We implement our experiments on a platform with one CPU, Intel Xeon 6248R (Cascade Lake), 3.0GHz, 24-core, and one GPU, NVIDIA A100 40GB GPU accelerator. For Synthetic Dataset Benchmarking in Section 4, the PCGSR algorithms produce 60 sampled expressions per second on average. For the materials science application in Section 5, PCGSR algorithms on average produce 20 sampled expressions per second.

First-principles DFT calculations of the test dataset were performed on a compute node consisting of two Intel Xeon 6248R (Cascade Lake) CPUs with a total of 48 cores and 384GB DDR4 memory. The initial supercell structure of 32-atom fcc copper was relaxed using VASP with conjugate gradient method in four steps within a total of 18 seconds. For each temperature, VASP-based molecular dynamics simulations in the NVT ensemble were run for 6,000 steps for a total of 900 minutes.

## B  METHODOLOGICAL DETAILS

### B.1  TRANSPLANTATION AND CONSTANT OPTIMIZATION

**Transplantation:** Inspired by the cross-over mechanism in Genetic Programming (GP), we introduce the concept of function modularity to break down complex problems into smaller sub-problems (Udrescu & Tegmark, 2020; Sun et al., 2022). Leveraging a divide-and-conquer heuristic search strategy, we store promising expressions with lower complexity than the current sampling average in a budget set. These stored expressions can then be directly inserted into newly generated expressions as a single operator, which we refer to as the "transplantation" operator, denoted by $trans$. The $trans$ operator functions as a feature that PCGSR can choose during exploration. When selected, PCGSR randomly picks a subset of expressions from the budget and uses them to generate new expressions, selecting the best one to calculate the action value of the $trans$ operator. By default, PCGSR enables the $trans$ operator after half of the total epochs, with the budget set to store the top $n$ models that have a complexity lower than half of the maximum complexity.

**Constants Optimization:** Constants play a crucial role in symbolic regression, not only for maintaining unit consistency in real-world scientific problems but also for optimizing regression accuracy. In our approach, constants are introduced into expressions via the $const$ operator, functioning as a feature operator. While enabling constant optimization enhances the accuracy of the regression, it significantly increases the computational cost of evaluating an expression. To mitigate this, all constants within an expression are optimized only once per evaluation using a non-linear regression method, specifically the BFGS optimizer, as described in Petersen et al. (2020).

### B.2  SG REPRESENTATIONS FOR SGNN AND MCTS DETAILS

To leverage the Symbolic Graph (SG) representation $G$ as the input for SGNN encoding, we can directly feed $G$ into SGNN due to the graph convolutional network (GCN)'s inherent aggregation and message-passing mechanisms, which naturally preserve the symmetries encoded in SG. However, to use SG as a state in Monte-Carlo Tree Search (MCTS), we first convert $G$ into a canonical form by sorting the nodes. Next, we modify the upper triangle of the sorted adjacency matrix by replacing its elements with the corresponding edge features, where "0" denotes a disconnected edge. The diagonal elements are replaced by the node features. Finally, we output this modified sorted adjacency matrix in its Voigt form, ensuring the representation of states while preserving symmetries.

### B.3  SGNN MESSAGE PASSING DETAILS

We adopt a similar graph convolutional network (GCN) structure as in Xie & Grossman (2018) for SGNN, which adopts a node updating function in the form:

$$v'_i = v_i + \sum_j \sigma(z_{ij} W_f + b_f) \odot g(z_{ij} W_s + b_s) \tag{6}$$

where $z_{ij} = v_i \oplus v_j \oplus \{e_{ij}\}$. $v_i$ is the node messages (also the node feature embedding for the first input layer) and $v_i'$ is the updated node messages for node $i$. $e_{ij}$ represents the edge embedding for edge features. $W_f$ and $b_f$ are the convolution weight matrices and convolution weight bias. $W_s$ and $b_s$ are self-weight matrices and self-weight biases for the attention mechanism. $\sigma$ and $g$ are soft plus activation functions and $\odot$ represent the element-wise multiplication. To output the prior distribution **P**, we use a multilayer perceptron (MLP) with softmax activation on the node messages, while a separate MLP with the softplus activation is employed to produce the global readout on node messages for predicted rewards.

### B.4 SGNN Encoding with Coefficient Control

In Section 3.3, we implement an SGNN-guided MCTS simulation policy controlled by the coefficient $\epsilon$, which governs the balance between exploration and exploitation. When $\epsilon = 1$, PCGSR performs random roll-outs like naive MCTS, while $\epsilon = 0$ means PCGSR exclusively uses SGNN for simulations. This parameter is designed to facilitate a smooth start-up in PCGSR training. Initially, SGNN lacks sufficient information about the dataset, making random roll-outs more effective. Therefore, for the first third of the total epochs, we set $\epsilon = 1$ to encourage exploration. As training progresses and SGNN becomes more informed, $\epsilon$ is gradually reduced to 0.2, allowing SGNN to guide more simulations and prioritize exploitation.

### B.5 Algorithms

**Algorithm 1** summarizes the pseudo-code for our PCGSR. The core of the algorithm is the repeated sampling of expression trajectories $\tau$ until either the desired solution is found or the maximum number of iterations is reached. At each trajectory step $t$, a batch of MCTS with index $k$ and a maximum batch size $B$ is executed, following the four steps for each MCTS iteration.

---

**Algorithm 1** Physics-Constrained Graph Symbolic Regression

> **Input:** operator and feature dictionary $\mathcal{Q}$, Batch size $B$, maximum complexity $H$
> **repeat**
>    **for** $t = 1$ **to** $H$ **do**
>       **for** $k = 1$ **to** $B$ **do**
>          Do one MCTS(selection, expansion, simulation, backpropagation) with $\pi_\theta$
>       **end for**
>       Collect $N(s, a)$ from MCTS and calculate $\pi_M$
>       Sample $\phi_t \sim \pi_M(s_t)$, $\phi_t \in \mathcal{Q}$
>       Expand $G_t$ with $\phi_t$, update $\mathcal{V}$ and $\mathcal{E}$
>       **if** $G_t$ is complete **then**
>          **break**
>       **end if**
>    **end for**
>    Calculate $R(\tau)$, record $\pi_\theta(s_t)$ and $\pi_M(s_t)$
>    Calculate the loss $l$
>    Update $\pi_\theta$ according to $l$
> **until** Optimal $f$ is found

---

## C  Further Experimental Details

### C.1  PCGSR Experimental Settings

The Feynman dataset is a widely accepted dataset adopted by the prevailing SR benchmarking framework, SRBench. It is a synthetic dataset consisting of 100 physics-inspired equations derived from the Feynman Lectures on Physics and 20 more challenging tasks (Matsubara et al., 2024). For the benchmarking with the Feynman dataset, we only consider equations involving up to 10 features, resulting in 119 valid Feynman equations. Each of the equations is conducted three times with different random seeds. Each of the benchmarking methods is restricted to 500,000 evaluations, a run-time budget of 24 hours, and a maximum complexity of 50 for the expressions. We follow

the same dictionary set (available operators and features) and other default experimental settings in SRBench. GP and AI Feynman 2.0 utilize default hyperparameters from SRBench while SPL and NGGP inherit default hyperparameters from their open-access code.

For the additional hyperparameters used in PCGSR, we apply consistent settings across all experiments in this paper. The MCTS batch size is set to $B = 1000$. The SGNN controlling coefficient $\epsilon$ starts at 1.0 for the first 33% of the total epochs and then linearly decays to 0.2 by the 66% epoch mark. For the transplantation process, we maintain a budget size of 100 for stored expressions and sample 10 expressions each time transplantation is employed.

## C.2 NYUGEN'S BENCHMARK RESULTS

| Benchmark | Expression | GP | NGGP | SPL | PCGSR |
|---|---|---|---|---|---|
| Nguyen-1 | $x^3 + x^2 + x$ | 99% | 100% | 100% | 100% |
| Nguyen-2 | $x^4 + x^3 + x^2 + x$ | 90% | 100% | 100% | 100% |
| Nguyen-3 | $x^5 + x^4 + x^3 + x^2 + x$ | 34% | 100% | 100% | 100% |
| Nguyen-4 | $x^6 + x^5 + x^4 + x^3 + x^2 + x$ | 54% | 100% | 99% | 100% |
| Nguyen-5 | $\sin(x^2)\cos(x) - 1$ | 12% | 80% | 95% | 100% |
| Nguyen-6 | $\sin(x) + \sin(x + x^2)$ | 11% | 100% | 100% | 100% |
| Nguyen-7 | $\log(x+1) + \log(x^2+1)$ | 17% | 100% | 100% | 96% |
| Nguyen-8 | $\sqrt{x}$ | 100% | 100% | 100% | 100% |
| Nguyen-9 | $\sin(x) + \sin(y^2)$ | 76% | 100% | 100% | 100% |
| Nguyen-10 | $2\sin(x)\cos(y)$ | 86% | 100% | 100% | 100% |
| Nguyen-11 | $x^y$ | 13% | 100% | 100% | 100% |
| Nguyen-12 | $x^4 - x^3 + \frac{1}{2}y^2 - y$ | 0% | 4% | 28% | 52% |
| Nguyen-1$^c$ | $3.39x^3 + 2.12x^2 + 1.78x$ | 0% | 100% | 100% | 100% |
| Nguyen-2$^c$ | $0.48x^4 + 3.39x^3 + 2.12x^2 + 1.78x$ | 0% | 100% | 94% | 100% |
| Nguyen-5$^c$ | $\sin(x^2)\cos(x) - 0.75$ | 1% | 98% | 95% | 100% |
| Nguyen-8$^c$ | $\sqrt{1.23x}$ | 56% | 100% | 100% | 100% |
| Nguyen-9$^c$ | $\sin(1.5x) + \sin(0.5y^2)$ | 0% | 90% | 96% | 94% |
| | Average Recovery Rate | 38.2% | 92.5% | 94.5% | **96.6%** |

Table 4: Recovery Rate of PCGSR and other baseline models benchmarked on Nguyen's SR benchmark dataset and Nguyen's SR benchmark with constants dataset (marked with upper index $c$). The recovery rate is the ratio of ground-truth equivalent solutions in mathematics to the total of parallel experiments for the same equation.

Our proposed PCGSR method is also evaluated in Nguyen's SR benchmark dataset (Uy et al., 2011) and Nguyen's SR benchmark with constants dataset (Petersen et al., 2020), two widely adopted synthetic datasets for evaluating the performance and robustness of various SR algorithms. The results, presented in Table 4, include the same baseline models and hyperparameter settings from Table 1.

The objective of this set of experiments is to find an expression $f(\cdot)$ that best fits the corresponding input features $(x, y)$ to the target output in these synthetic datasets. The "Expression" column in Table 4 includes the ground-truth expressions used to generate synthetic data. In this set of experiments, we adopt results of GP, NGGP and SPL from Sun et al. (2022) and follow the same settings to implement PCGSR, which uses a dictionary set $\mathcal{Q}_0 = \{\times, +, -, \div, sin, cos, exp, log, x\}$ for the benchmark Ngugen-1 to Ngugen-8, and $\mathcal{Q}_1 = \mathcal{Q}_0 \cup \{y\}$ for the benchmark Ngugen-9 to Ngugen-12. It is worth noticing that for the benchmark Nguyen-8, $\sqrt{x}$ can be recovered from $\exp\left(\frac{x}{x+x}\log(x)\right)$. For Nguyen-10, $x^y$ can be recovered from $\exp(y\log(x))$. For Nguyen-7 and Nguyen-10, they can also be recovered from $\log(x^3 + x^2 + x + 1)$ and $\sin(x + y)$. The maximum complexity $H$ for each expression is set to 35. The batch size $B$ is set to 1,000 for MCTS simulations, and the maximum number of episodes is capped at 1,000, resulting in a total search space of up to 1 million expressions for MCTS and PCGSR. The training and testing datasets are divided equally, with 20 randomly generated data points for each.

Table 4 shows that PCGSR outperforms competing SR approaches in most tasks, particularly in complex tasks like Nguyen-12. This advantage is primarily due to PCGSR's effective management of polynomial terms, facilitated by the symmetries captured by SG, which compresses the search space more efficiently than the expression tree or other representations utilized by alternative methods, highlighting its considerable potential in the domain of Symbolic Regression.

### C.3 ABLATION STUDY SETTINGS

The benchmark problems in the Feynman dataset with features of two dimensions were chosen across different complexities, from easy to hard, to fairly compare efficiency and accuracy in different scenarios. This includes one expression of complexity 2, three expressions of complexity 3, one expression of complexity 4, one expression with complexity 7, one expression with complexity 9, and one expression with complexity 12

## D MATERIALS SCIENCE APPLICATION

### D.1 PROBLEM BACKGROUNDS

The physical world is composed of numerous ions and electrons, governed by quantum mechanics, often referred to as many-body problems. While density functional theory (DFT) can simulate simple materials with few ions and electrons, extending DFT to larger systems is challenging. One of the goal relevant to the physical properties of materials is to find an analytical approximation that relates DFT-calculated potential energy to the interatomic distance (i.e. pairwise distances) between atoms, adhering to Pauli's exclusion principle. Using copper (Cu) crystal as an example, we aim to identify analytical relationships between interatomic distances and total potential energy. This functional form is crucial for materials science as it enables large-scale molecular dynamics simulations to study materials' mechanical, thermal, and kinetic properties and explain fundamental physical mechanisms at the atomic level.

The following provides some explanations for domain-specific terms:

- Molecular dynamics (MD) simulates the structure and properties of materials under constant or varying environments (e.g. temperature or mechanical strain). In MD simulations, the atomic forces are computed from the derivatives of interatomic potential energy functions with respect to the atomic distance placement, which then moves the atoms based on Newton's second law. The updated atomic positions lead to new interatomic potential energy. The calculations are performed iteratively until a desired number of time steps. Statistics of total potential energy, kinetic energy, atomic forces, and stresses, etc. provide a systematic understanding and quantitative estimate of the physical properties of materials. MD simulations are often performed for systems with hundreds to millions of atoms, making them computationally expensive. This necessitates machine learning algorithms like those employed in our study. Analytical expressions with low complexity and high accuracy are particularly valuable for large-scale MD simulations.

- In MD simulations, the system state can be specified using ensembles like NVT (constant number of atoms, volume, and temperature), allowing other properties such as pressure and chemical potential to vary.

- Atomistic simulations based on machine learning-derived potential energy functions use atomic species and interatomic forces to determine material properties. This method is efficient but requires accurate fitting from first-principles / quantum mechanics-based calculations, such as density functional theory (DFT).

- Periodic boundary conditions (PBCs) enable the modeling of infinite or effectively infinite systems by repeating a "unit cell" in all directions. This unit cell represents the smallest section of the structure that can be repeated to create the correct crystal structure, allowing large systems to be modeled efficiently without losing structural or symmetrical integrity.

## D.2 PRATICAL USAGE OF PCGSR'S SOLUTION

- **Framework Generality**: PCGSR is not limited by system size or atom types. The 32-atom copper dataset is chosen to serve as a benchmark to compare PCGSR with prior work and demonstrate its advantages. The derived potential energy function can be applied to larger systems.

- **Dataset Relevance**: The 32-atom copper dataset is representative of materials studies, consistent with the average system size ($\sim$ 31 atoms/structure) of the widely used MPtraj dataset (Deng et al., 2023).

- **Scalability**: While DFT methods are limited to simulating small systems (1–1000 atoms) over very short timescales (a few picoseconds), our method enables large-scale simulations (millions of atoms) over extended timescales (microseconds)- which can then help determine many physical properties of materials that are not possible for quantum chemistry methods such as DFT. The analytical function from PCGSR can be applied to study real material problems with sizes far beyond 32 atoms for long-time dynamics that are completely inaccessible to DFT or other quantum chemistry methods – which is the key purpose of developing force field based on accurate analytical potential energy functions from the PCGSR method. As a result, one can simulate the mechanical deformation process of single or polycrystalline copper consisting of millions of copper atoms using large-scale molecular dynamics simulations with the potential energy function developed here. For example, studying the novel stacking of copper in the incubation period of crystallization (Liu et al., 2023) requires the simulation of millions of copper atoms.

- **DFT-Level Accuracy**: The dataset is derived from DFT calculations, enabling DFT-level accuracy but without explicit electron degree of freedom - that's the whole purpose of developing potential energy function (or, machine learning force field) from quantum chemistry datasets such as DFT. The high accuracy, the analytical nature, and the proper limiting trend at short bond length make this method and the derived potential energy function particularly useful.

## D.3 EXPERIMENTAL SETTINGS

**Dataset Description** The 32-atom copper dataset consists of 150 snapshots generated by the Vienna Ab initio Simulation Package (VASP), a first-principles DFT package. Each sample includes total formation energy $E$ as the target and a crystal structure of copper as the feature. To map a copper crystal structure to $E$, we first convert the 3D coordinates of the structure into pairwise distances $r$ between atoms within the unit cell in the crystal under the periodic boundary condition (PBC) (Makov & Payne, 1995), then use the surrogate model as the interatomic potential energy function $f(\cdot)$ to obtain $E = f(r)$. During the conversion, we only consider pairwise distances within a cutoff range of $r < r_{\text{cutoff}} = 5\text{Å}$.

**SR Settings** In the symbolic regression configurations, we utilize a dictionary set $\mathcal{Q} = \{+, -, \times, \div, \wedge, \sum, exp, \text{const}, r\}$, where "$\sum$" is used for the summation-based aggregation operation, "$\text{const}$" denotes constants optimized through non-linear regression. The maximum complexity $H$ is set at 35, and the batch size $B$ is 1,000, with 5,000 episodes allocated for both MCTS and PCGSR methods. For the black-box CGCNN method, we adhere to the default hyperparameter settings with a maximum of 5,000 epochs. For the train-test split, we follow Hernandez et al. (2019) with 50% for training and 50% for validation.

## D.4 ADDITIONAL FIGURES

Figure 4 plots the data fitting for SR solution models in Table 3.

## D.5 EXTRA EXPLANATIONS FOR PHYSICS CONSTRAINTS

Here we offer detailed explanations and examples for some physics constraints defined in Section 5.2:

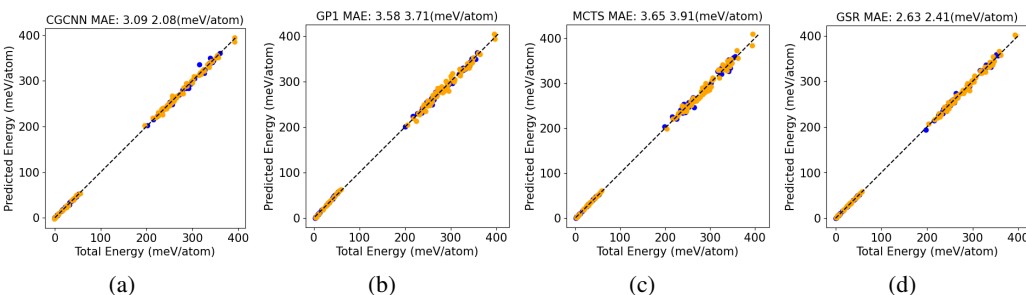

Figure 4: Training (orange dots) and Testing (blue dots) MAEs of the formation energy predictions by CGCNN (a), GP1 (b), MCTS-SG (c) and PCGSR (d) on DFT dynamics simulations of FCC copper. The dashed lines mark the identity mapping. The top captions of the plots also include testing MAE (left) and training MAE (right).

- **(2) Unit Consistency:** For a specific example of the unit consistency constraint, the expression of $(r^2 + r)$ is not physically meaningful, as $r$ has a length unit Å. and we would have an inconsistent unit ($Å^2$ +Å) from the expression. But a constant $c$ can include unit Å so that we have a consistent calculation with $r^2 + c * r$. The introduced $const$ can ensure that the final output unit aligns with the target $E$'s unit as electron-volt ($eV$; $1\ eV = 1.6 \times 10^{-19}$ Joule) from the input $r$'s unit Angstrom (Å; 1Å= $10^{-10}m$).

- **(3) Electrostatic Repulsion:** When the distance between two atoms approaches zero, their electron wavefunctions start to overlap significantly which is excluded by the Pauli's principle. Thus, the atomic orbitals hybridize and form molecular orbitals with bonding and antibonding characters and electron density is thus pushed away from nuclei, leaving the repulsive nucleus-nucleus electrostatic interaction being the dominant one and approaching infinite potential energy at very short distance.

### D.6 PHYSICAL MEANING

Figure 3 presents the potential energy curves as a function of the interatomic distance (bond length) for the expressions generated by the GP1, GP2, MCTS-SG, and PCGSR in Table 3. These curves demonstrate whether the fitted models satisfy electrostatic repulsion at zero bond length. Figures 3a and 3b show that GP1 and GP2 violate the constraint due to the finite value at zero bond length ($r$=0). In contrast, Figures 3c and 3d show that MCTS-SG and PCGSR yield infinite potential energy at $r$=0, satisfying the principle. Such a constraint is crucial not only for the underlying physics but also for practical applications in materials system simulations. The missing divergence at $r = 0$ may cause atoms to collapse to each other during molecular dynamics simulations and yield incorrect results.

Additionally, Figures 3a and 3b reveal two critical issues with GP1 and GP2: (i) a discontinuity at $r$=1Å, and (ii) an infinite value when $r$ approaches 5Å. These issues lead to incorrect energy and force predictions, forcing atoms to remain unrealistically close together.

### D.7 GENERATION OF NEW DATASET FOR TESTING MODEL TRANSFERABILITY

To test the transferability of the models, we generated a test dataset from DFT calculations using the VASP package (Kresse & Furthmüller, 1996) where the projector augmented wave method was applied to treat core electrons (Blöchl, 1994). We used the Perdew-Burke-Ernzerhof (PBE) form of exchange-correlation energy functional within the generalized gradient approximation (Perdew et al., 1996) and a Monkhorst-Pack $k$-point sampling grid of $4 \times 4 \times 4$ (Monkhorst & Pack, 1976). A hydrostatic compression was applied to fcc copper with strain of -0.2 along all lattice vectors. We then performed first-principles molecular dynamics simulations in the NVT ensemble for total 6,000 steps with a time step of 3 fs at two different temperatures, *i.e.* 300 K and 1,400 K. For the NVT calculation at each temperature, we excluded the first 1,000 steps of the initial equilibration process, and extracted the atomic structures for every 100 steps from the remaining equilibrated 5,000 steps, which yields total 100 snapshots (or samples) to test model transferability.

The compressed dataset has the following physical meaning:

- **Scientific Context**: Understanding matter in extreme conditions such as high pressure and high temperature is an important and active subject of materials research. Copper is one of the systems of particular interest. As shown in McCoy et al., 2017 [3], the density in experiments reaches $\sim$18 g/cm$^3$, double the density at the standard condition of 8.95 g/cm$^3$, that is, the volume contraction by 50%. Another example is done by Fratanduono et al. (2020) at the National Ignition Facility (NIF) at the U.S. Lawrence Livermore National Laboratory (LLNL), in which the solid copper was even compressed to $\sim$28 g/cm$^3$ at terapascal conditions, corresponding to $\sim$67% volume contraction.

- **Practical Necessity** Beyond scientific motivation, another key motivation to apply large compression is to provide a more accurate trend away from equilibrium towards $r \rightarrow 0$. This is particularly important as the machine learning interatomic potentials or machine learning force fields very often have wrong limiting behavior. When they were applied to simulating long-time dynamics at high temperature or high pressure, there will be an increasing probability of "direct crossing or fusion" of atoms which are pure artifacts due to the wrong limiting trend, consequently, the results can be completely nonphysical and wrong.

