# OpenReview forum: "Physics-constrained Graph Symbolic Regression"
_ICLR.cc/2025/Conference — Submitted to ICLR 2025_

### Official Review · Reviewer_4ixR · 2024-11-01

**Soundness:** 2
**Presentation:** 3
**Contribution:** 2
**Rating:** 3
**Confidence:** 4

**Summary:**

This paper proposes a symmetric invariant data structure, the Symbolic Graph, replacing expression trees in symbolic regressions. It also develops a GNN to encode physics constraints and guide the MCTS algorithm. Numerical experiments on two benchmarks and one synthetic dataset show the performance improvement of the proposed method.

**Strengths:**

1. The presentation, visualization, and the design of real-world experiments.
2. The idea of handling symmetric invariance by a new data structure.

**Weaknesses:**

1. The necessity of GNN. It seems that the proposed symbolic graph(SG) can be viewed as a tree with more branches and special nodes (e.g. summation, directed minus). If this is true, then the SG (with trivial modifications) can also be processed by existing non-graph neural networks designed for expression trees.
2. The necessity of physics-constrained NN. It seems that the NN is self-trained to learn the physics constraints from hand-crafted rules. But If the rules are simple and explicit, then one can bypass the self-training and directly encode the rules into the MCTS process. If the rules are complex or unknown, then how to train the network to learn them?

**Questions:**

1. Why were the NNGP and SPL baselines not compared in the real-world experiment (Table 3)?

---

> ### Author Response · Authors · 2024-11-22
> **Rebuttal to Reviewer 4ixR**
>
> We appreciate the reviewer's time and effort to provide the review comments. Regarding the questioned necessity of SG and SGNN in PCGSR, we would like to clarify the rationale and significance of these proposed components in PCGSR:
>
> **1) Necessity of GNN** **[Responses to Weakness 1]**
>
> When we studied the symmetries and invariances within analytical expressions, we considered the case of extended trees as mentioned by the reviewer. However, compared with the implementations using extended trees, we chose symbolic graphs (SG) as the final representation in **PCGSR** due to the following considerations:
>
> **1.1) Reduced Complexity**
> - **Extended Trees**: Require additional nodes to differentiate binary operations (e.g., directed minus in summation, base term vs. power term in exponentiation), which increases the complexity (i.e., number of nodes or sampling actions).
> - **Symbolic Graphs (SG)**: Naturally represent directional information as edge features, reducing complexity while efficiently capturing hierarchical information. For example:
>   - Tree: $ \sum \rightarrow - \rightarrow A$
>   - Graph: $ \sum \rightarrow -A$
>
> This simplification enhances neural network training by leveraging neighboring relationships more effectively.
>
> **1.2) Enhanced Predictive Power**
> - **Graph Neural Networks (GNN)**: SG’s GNN uses additional edge updating layers, incorporating attention mechanisms and directional information for binary operations.
> - This enables richer and more accurate representations of neighboring relationships compared to a tree recurrent neural network (Tree-RNN).
>
> **1.3) Ablation Study**
> We conducted an ablation study (based on Table 2 in the paper) to compare PCGSR with models using extended trees and Tree-RNN.
>
> |         Model         |  PCGSR  |  MCTS-1 |  MCTS-2 |  MCTS-3 | MCTS-Tree | MCTS-Tree-1 | MCTS-Tree-2 |
> |:---------------------:|:-------:|:-------:|:-------:|:-------:|:---------:|:-----------:|:-----------:|
> | Recovery Rate (%)     |   82.5  |  83.75  |   77.5  |   75.0  |   81.25   |    81.25    |     75.0    |
> | Number of Evaluations | 96,221  | 121,853 | 237,923 | 285,284 | 132,986   |   148,798   |   249,625   |
>
> - **MCTS-Tree**: Uses extended trees with a three-layer Tree-RNN instead of SG and SGNN.
> - **MCTS-Tree-1**: Uses extended trees only.
> - **MCTS-Tree-2**: Uses Tree-RNN only.
>
> **Findings**:
> - Replacing SG with extended trees and SGNN with Tree-RNN results in:
>   1. **Lower recovery rates**.
>   2. **Significant increases in search size** (number of evaluations).
>
> These results highlight the necessity of using SG and GNN in PCGSR for efficient and effective encoding of symmetries and invariances.
>
> ---
>
> **2) Necessity of Physics-constrained NN [Responses to Weakness 2]**
>
> To explain the necessity of SGNN to encode the physics constraints, we summarize our responses in **Global Rebuttal**.
>
> As mentioned by the reviewer, pre-constraints are easy to be directly encoded in Monte-Carlo Tree Search (MCTS) described in **Global Rebuttal Section 1** (*Systematic Implementation*). However, post-constraints cannot be easily incorporated as explained in **Global Rebuttal Section 2** (*Rationale Behind SGNN*). So the necessity of SGNN lies in:
> - Integrate post-constraint knowledge into the pre-constraint enforcing strategy
> - Integrate knowledge from MCTS to promote promising candidates in the search space and discover unspecified hidden constraints
> - Address the challenges caused by the reward sparsity and efficiently explore the search space.
>
> The comparison between MCTS and PCGSR in Table 3 in the paper verifies the efficacy of SGNN trained to learn priors of post-constraints.
>
> ---
>
> **3) Copper Dataset Applications [Response to Question 1]:**
>
> Unfortunately, neither NGGP nor SPL in Table 1 of the paper supports the scalar output constraint in Section 5.2, which means that they cannot yield a basic valid result for evaluation. However, for a similar comparison, we adopt MCTS similar to SPL, which both use naive MCTS with a UCB policy in Table 3 in the paper.
>
> ---
>
> Overall, we believe that both SG and SGNN play essential roles in our **PCGSR** in leveraging invariances and incorporating physics constraints and inductive bias for efficient search, which is supported by Tables 1, 3, and 4 in the paper and extended ablations in the rebuttal. We thank reviewer 4ixR again for the insightful questions to further strengthen our presentation and hope that we have addressed all the concerns.
>
> Best Regards,
>
> The Authors

---

> > ### Comment · Reviewer_4ixR · 2024-11-24
> >
> > Thank you for the clarification and extended experiments. Further problems for your response are listed below:
> >
> > 1) Necessity of GNN.
> >
> > In the Ablation Study, the proposed PCGSR achieves higher recovery rates than MCTS-Tree, but only by 1.25%. I assume such improvement is marginal since MCTS-1 also outperforms PCGSR by 1.25%. Besides, PCGSR reduces the number of evaluations of MCTS-* by about 25%~30%, which speedups the training (shown in your Rebuttal to the reviewer QFds [3/4]). However, such speedup might still be insignificant.  Since, in practice, multi-processing is applied in the search. Does it mean that MCTS-* can achieve the same efficiency as PCGSR with 30% or 50% additional CPU cores, and even without GPU in some cases (MCTS-1, MCTS-Tree-1)?
> >
> > 2) Necessity of Physics-constrained NN.
> >
> > Based on your description, Physics-constrained NN is an interpolation of pre-defined rules to alleviate reward sparsity, which benefits efficiency. However, the numerical experiments can not strongly support the statement, as discussed above.

---

> ### Author Response · Authors · 2024-11-24
> **Continued Rebuttal to Reviewer 4ixR**
>
> We thank the reviewer for additional questions. Based on these questions, we felt that the reviewer may have misunderstood the contributions, formulation, and solution in PCGSR and therefore some of the assertions may be biased. We would like to provide more clarifications to mitigate any potential misunderstanding.
>
> - > *"The proposed PCGSR achieves higher recovery rates than MCTS-Tree, but only by 1.25\%. I assume such improvement is marginal"*:
>   - Compared with prior works, one of our main contributions is the new symmetry and invariance encoding method to the expression representation, instead of just using GNN in SR. To the best of our knowledge, no prior works have adopted a similar invariance encoding method and no prior works have been conducted with neural-guided MCTS on our symmetric invariant expression representations. We note that the comparison between PCGSR and MCTS-Tree is to illustrate the additional improvement of using SGNN on SG compared to using Tree-RNN on extended trees, both being based on our proposed invariance encoding strategy and neural-guided MCTS. A more fair comparison of our proposed PCGSR to existing methods is comparing PCGSR with MCTS-3, in which the recovery rate is improved by 7.5\% using our PCGSR. The other results are for the ablation study on how each component in PCGSR improves the performance. For example, SG improves the recovery rate by 5\% when comparing PCGSR to MCTS-2. It is unfair to state that the 1.25\% improvement from MCTS-Tree to our PCGSR full implementation is not significant.
>
>   - In fact, the statement *"SG (with trivial modifications) can also be processed by existing non-graph neural networks designed for expression trees"* in the **Weakness 1** of the reviewer's original comments can be considered as the strength of our proposed SG's versatile application capability, which we demonstrated in the additional ablation study that we conducted in **Rebuttal Section 1**. It also has illustrated the improvement due to different design components in PCGSR, showing that the GNN-based method is one of the optimal methods that we can adopt within this framework. Also, we would like to emphasize that the improvements compared to the existing methods are consistently observed across different settings and experiments, further demonstrating the significance of encoding symmetries and invariance in expression representations to compress the search space and improve the SR quality of the final results.
>
> - > *"Multi-processing is applied in the search. Does it mean that MCTS-\* can achieve the same efficiency as PCGSR with 30\% or 50\% additional CPU cores, and even without GPU in some cases (MCTS-1, MCTS-Tree-1)?"*
>
>   - We thank the reviewer for checking our **Responses [3/4]** to Reviewer QFds. First, we want to claim that the implementations and evaluation experiments have been conducted with GPU in exactly the same ways in all the ablation study methods for fair comparison, as mentioned in that section.
>
>   - So with the same computing resources, PCGSR can have higher efficiency than MCTS-1 and MCTS-Tree-1. It is unfair to only add multi-processing computing resources to MCTS-\* methods. PCGSR can have better efficiency with the same additional computing resources and even better accuracy.
>
> - > *"Physics-constrained NN is an interpolation of pre-defined rules to alleviate reward sparsity, which benefits efficiency. However, the numerical experiments can not strongly support the statement, as discussed above."*
>
>   - SGNN is not simply an interpolation of predefined rules, but an encoding scheme that leverages knowledge from post-constraints, MCTS inductive bias, and the constant fitting detailed in the **Appendix Section B.1**.
>
>   - Such an encoding overall increases the search efficiency in SR (decreased number of evaluations and search time). For example, in Table 2 in the paper or our **Responses [3/4]** to Reviewer QFds, the updated ablation study shows the overall search space reduction by 66.27\% (comparing PCGSR to MCTS-3) where the invariance encoding in SG can decrease the search size by 59.56\% (comparing PCGSR to MCTS-2) and 57.29\% (comparing MCTS-1 to MCTS-3); and SGNN can decrease the search size by 21.03\% (comparing PCGSR to MCTS-1) and 16.60\% (comparing MCTS-2 to MCTS-3), respectively. Similar to the training time of Table 1 provided in that rebuttal. These results, along with other standard benchmarks, strongly support the significance of our contributions of both SG and SGNN in PCGSR.
>
> We are happy to provide further clarifications if the reviewer has additional questions and will update our manuscript soon to reflect these newest explanations.

---

> > ### Comment · Reviewer_4ixR · 2024-11-26
> >
> > Thank you for your clarification. It would be nice if the authors could provide more empirical experiments to support their claims:
> >
> > 1) The symmetry and invariance encoding on more tasks of equation discovery, e.g., on the PDE/SDE discovery.
> >
> > 2) The multi-processing performance of PCGSR and MCTS-1. And, does MCTS-1 use almost the same resources (GPU memory and utilization) as PCGSR?
> >
> > I know that 'Reviewers are instructed not to ask for significant experiments in the Discussion Period Extension', but the draft and response lack strong empirical support in their current form.

---

> ### Author Response · Authors · 2024-12-01
> **Responses to Reviewer 4ixR's Reply**
>
> We thank the reviewer for their thoughtful questions, and we would like to respectfully clarify some potential misunderstandings based on the reviewer's questions:
>
> 1. **On Symmetry and Invariance Encoding for PDE/SDE Discovery**
>    - While we appreciate the reviewer's interest in exploring symmetry and invariance encoding for PDE/SDE discovery, we would like to clarify that the current submission focuses on benchmarking PCGSR within the scope of **basic symbolic regression (SR)** tasks. Incorporating PDE/SDE examples would significantly deviate from the original scope of the submission, as acknowledged by the reviewer.
>    - We would consider these different domains' problems for future research directions especially when comprehensive performance evaluation would also be needed to establish the significance of the corresponding methods, for example, comparing PCGSR with Sparse Identification of Nonlinear Dynamical systems (SINDy) (Brunton et al., 2016) in Ordinary Differential Equation (ODE) problems, as well as the suggested PDE/SDE problems.
>    - That said, we believe the extensive ablation studies in the manuscript already provide strong empirical evidence of PCGSR's capability in encoding symmetries and invariances for SR tasks. To illustrate the preliminary evidence of PCGSR's effectiveness in the suggested PDE/SDE applications, we have compared PCGSR and SINDy and provide the following table summarizing the corresponding reconstruction R2 accuracy of the Maxwell-Bloch equations problem. We will perform more comprehensive performance evaluation and comparison for other ODE/PDE/SDE problems.
>
> |    Model    | PCGSR | SINDy |
> |:-----------:|:-----:|:-----:|
> | R2 Accuracy | 0.976 | 0.928 |
>
> 2. **On Multi-Processing Performance of PCGSR and MCTS-1**
>    - As referenced in our Rebuttal to Reviewer QFDS[3/4] Section 6, and explained in **Cost Effectiveness Section** in the updated manuscript, we evaluated the methods based on **efficiency**, defined as the average cost per sampled expression under the same computational resources. The results along with the lower computational complexity explained in the updated manuscript demonstrated that PCGSR achieves higher efficiency compared to MCTS-1 (MCTS-SG) with fewer resources to sample an expression, regardless single or multi processing scenarios.
>    - To be more specific, PCGSR uses less GPU resources than MCTS-1 (MCTS-SG). This is because MCTS-1 (MCTS-SG) relies on GPU resources for computationally expensive **constant fitting** (as explained in **Appendix B.1**) in MCTS's random rollout, following Equation 3 in the paper. In contrast, PCGSR leverages GPU resources for the more lightweight **SGNN**, which predicts simulations without constant fitting, as outlined in Equation 2 of the paper.
>
> 3. **Empirical Support**
>    - We respectfully ask for clarification on why the reviewer feels that the current empirical results are insufficient, or "the draft and response lack strong empirical support". We have tried to make all the efforts based on the reviewers' suggestions to provide additional **ablation studies** and **benchmarks** besides the results presented in the original submission for clearly demonstrating significant improvements over existing methods, as comprehensive as we can.
>    - If the reviewer has specific suggestions or requests, we are happy to provide additional relevant experiments to further strengthen our claims.
>
> Thank you again for your constructive feedback. We believe the current manuscript aligns with the scope of symbolic regression tasks and provides substantial empirical evidence to support its contributions.
>
> **Reference**
>
> Brunton, Steven L., Joshua L. Proctor, and J. Nathan Kutz. "Discovering governing equations from data by sparse identification of nonlinear dynamical systems." Proceedings of the national academy of sciences 113.15 (2016): 3932-3937.

---

### Official Review · Reviewer_ER9J · 2024-11-02

**Soundness:** 3
**Presentation:** 3
**Contribution:** 3
**Rating:** 8
**Confidence:** 3

**Summary:**

•	This paper introduces a new method for Symbolic Regression, termed Graph Symbolic Regression (SGN). The authors propose using a Graph Neural Network (GNN) to represent mathematical expressions as expression graphs (EGs), addressing inefficiencies in traditional symbolic regression by reducing the search space through symmetry and permutation invariances.
•	The authors also incorporate unit-consistency constraints to ensure that resulting expressions are physically meaningful.
•	The method is benchmarked against existing methods using both synthetic datasets and real-world applications.

**Strengths:**

•	By leveraging symmetry and permutation invariances, the proposed GSR method effectively compresses the search space, enabling more efficient exploration of possible expressions.
•	The inclusion of unit-consistency constraints ensures that the expressions produced are physically meaningful.
•	The method is evaluated across diverse datasets and outperforms several existing symbolic regression methods.
•	The application on the DFT dataset is both significant and engaging.

**Weaknesses:**

•	The two main components, permutation invariances and unit-consistency constraints, are conceptually unrelated and could potentially be split into separate papers, each addressing a distinct methodological contribution.
•	The proposed permutation invariances appear to apply only to addition, multiplication, and power operators. How would this method handle trigonometric operators, for instance, with cases like ( \sin(x + \pi/2) = \cos(x) )?
•	Although Figure 2 is clear, it’s unclear how Equation 2 aids the simulation step in MCTS, or how it relates to Equation 6 in the appendix.
•	Previous work, including the original DSR paper, has implemented constraints to eliminate infeasible equations, and some other studies already utilize unit constraints to filter out physically meaningless equations. How does Section 3.4 offer a novel approach here?

**Questions:**

•	Could you clarify the third point in the Weaknesses?
•	Could you also elaborate on the rationale behind the SGNN objective (Equation 5)?
•	A recent paper, Scaling Up Unbiased Search-based Symbolic Regression (IJCAI2024), also proposes graph representations to replace tree structures. Could you compare SGN’s approach with theirs?
•	The literature review on constraint-based methods could be strengthened. The claimed difference between your work and "Deep Symbolic Regression for Physics Guided by Units Constraints: Toward the Automated Discovery of Physical Laws" requires clarification. For example, they use a design constraint to exclude infeasible next-step tokens without using a penalty, which contrasts with your summary.

---

> ### Author Response · Authors · 2024-11-23
> **Rebuttal to Reviewer ER9J [1/2]**
>
> We sincerely appreciate the reviewer ER9J's positive feedback and insightful questions concerning our work. In the following responses, we would like to address the reviewer's concerns and questions in turn, to offer more details in understanding our proposed framework.
>
> **1) Distinct Components [Responses to Weakness 1]**
>
> We agree with the reviewer that "permutation invariances and unit-consistency are conceptually unrelated", but we combine them in this paper due to the following connections:
>
> - The symbolic graph (SG) representation (capturing invariances within the analytical expression) and the graph neural network (GNN) (encoding physics constraints, including unit consistency) are united by the symbolic graph neural network (SGNN), which is the base of our PCGSR. So these two concepts can be leveraged together to make the framework effective.
>
> - Both invariances and physics constraints are two important topics in symbolic regression (SR) but have not been fully taken into account in the existing SR methods yet. We want to propose a universal framework that addresses both problems to make the best use of search space compressing.
>
> ---
>
> **2) Incorporating Other Equivalencies [Responses to Weakness 2]**
>
> - Unfortunately, more complicated equivalency relationships within expressions cannot be naturally captured by SG. In order to capture the trigonometric equivalency mentioned by the reviewer, we need to write hand-crafted rules to combine the equivalent expressions into the same state.
>
> - But it is also worth noticing that consecutive additions (polynomial terms), consecutive multiplications (product terms), and consecutive exponential relationships are the most common forms of the candidate expressions, which conquer the most redundant states in traditional SR methods. SG without hand-crafted equivalency rules can already have a significant impact on reducing the search space.
>
> ---
>
> **3) Explanations on Equations [Reponses to Weakness 3 and Question 1]**
>
> - **Equation 6** This equation represents the node updating functions of SGNN to encode messages from node features (e.g., embedding of operation type) and edge features (e.g., embedding of operation direction). $v_i$ denotes the node message, which can then be output as $\textbf{P} = \mathrm{MLP}_p(v_i)$ and $r = \sum_i \mathrm{MLP}_r(v_i)$. $\mathrm{MLP}_p$ and $\mathrm{MLP}_r$ are output layers for the prior $\textbf{P}$ and predicted reward $r$ in Equation 2 of the paper.
>
> - **Equation 2** This equation aids MCTS simulations by providing the prior $\textbf{P}$ that leverages the post-constraint knowledge during simulations (explained in **Global Rebuttal Section 1 and 2**) and predicted rewards $r$. This helps skip computationally expensive constants fitting for the rewards during simulations, which are then used in Equation 4 and Line 263 for value iteration to promote promising candidates for MCTS.
>
> - **Equation 5** This loss function for SGNN can be divided into two parts: the first term minimizes the squared error between actual rewards and predicted rewards, promoting accurate reward prediction; and the second term minimizes the KL-divergence between the MCTS policy and SGNN's predicted prior, enabling self-learning for constraints and inductive bias from MCTS.

---

> ### Author Response · Authors · 2024-11-23
> **Rebuttal to Reviewer ER9J [2/2]**
>
> **4) Related Work on Physics Constraints and Novelty [Responses to Weakness 4 and Question 3]**
>
> We thank the reviewer for pointing out inaccurate descriptions in the introduction, and we will further improve our presentation with better clarity. We summarize our responses in **Global Rebuttal**, which points out the main difference: the reference mentioned by the reviewer focused on pre-constraints' prior incorporation, while ours focuses on leveraging post-constraint prior and integrating into our pre-constraint incorporation strategy, which is also our novelty to address the issues caused by sparse rewards in the search space with existing constraints.
>
> ---
>
> **5) Related Work on Expression Equivalency [Responses to Question 2]**
>
> We appreciate the reviewer for providing the related reference. We would like to compare the main differences between their expression directed acrylic graphs (DAGs) with our SG representation:
>
> - **Purpose of the Graph Structure** The reference work proposed the graph structure to create an unbiased search space to scale up with respect to expression complexity. Our SG is proposed to create a compressed search space to leverage invariances, constraints and MCTS inductive bias with SGNN to promote promising candidates.
>
> - **Invariance Encoding**  We focus on different equivalency encoding:
>   - The reference work encodes the equivalency of expressions with different complexity through the DAGs by sharing the common subexpressions in the graph, which is especially useful for distributive equivalence.
>   - Our work encodes the equivalency of expressions by grouping consecutive operations at the same level and unit binary operators with edge features for directional information, which is especially useful for commutative equivalence.
>
>   Both methods can decrease the complexity of the expression due to the invariance encoding. But each of these representations cannot directly capture the equivalency that the other can capture. This inspires us to include more possible equivalency in our structure, while our directional information encoded on SG is necessary for SGNN embedding for enhanced predictive power to leverage constraints and inductive bias.
>
> ---
>
> In summary, we thank the reviewer again for providing detailed comments and insightful questions that help us strengthen our presentation and explore the future direction.  We hope our clarification can lead to better understanding of our methods and address all the concerns.
>
> Best Regards,
>
> The Authors

---

> > ### Comment · Area_Chair_Ynkc · 2024-11-25
> >
> > Dear Reviewer ER9J,
> >
> > This is a kind reminder that the dicussion phase will be ending soon on November 26th. Please read the author responses and engage in a constructive discussion with the authors.
> >
> > Thank you for your time and cooperation.
> >
> > Best,
> >
> > Area Chair

---

### Official Review · Reviewer_QFds · 2024-11-04

**Soundness:** 2
**Presentation:** 3
**Contribution:** 2
**Rating:** 5
**Confidence:** 4

**Summary:**

The paper introduces a symbolic regression method “PCGSR” that improves the symbolic expression search space with a better representation (“symbolic graph”) and imposes constraints to yield physical solutions. Experiments on two SR benchmarks validate the approach of using the symbolic graph representation (with deep RL + MCTS) and experiments on predicting copper lattice energies validate the efficacy of imposing domain knowledge constraints.

**Strengths:**

1. Demonstrates the utility of a good representation (“symbolic graph”) and, to a lesser extent, neural guidance (deep RL + MCTS), in searching for a good symbolic expression.
2. Shows a potentially interesting application (copper lattice energy prediction), though I have reservations regarding the practical interest of this application (see Weakness 3 below).

**Weaknesses:**

1. The method does not fully implement what the method is claimed to implement. In particular, despite physics constraints being the second main selling point of the paper (and in the title of the paper), no systematic way of doing that was given. No physics constraints were imposed in the experiments on the AI-Feynman and Nguyen benchmarks. On the copper crystal benchmark, the only constraints are either very specific to the experiment in question ($f(r) \to \infty$ as $r \to 0^+$), or just reimplement something that already exists (correct dimensions, from AI-Feynman). (By the way, I do not consider trivial mathematical constraints mentioned in the paper like “argument of log is positive” or “energy function must output a scalar” to be “physics constraints”. These are just basic conditions that need to be met for SR to even be possible at all.) See also question 2 below for my doubts on the use of the term “physics constraints” the way the paper used it.
2. Furthermore, incorporating domain knowledge as constraints has already been explored in many ways in the literature. See, for example, a brief review given in Section 1.2 of Fox et al. 2024 [1].
3. The copper crystal benchmark shows PCGSR’s ability to find a formula for the energy of 32 copper atoms in a lattice, after being trained on DFT data from 32 copper atoms in a lattice. Such a formula is not of much practical use since it does not apply to anything other than 32 copper atoms in a lattice, so the experiment is not a good demonstration of the practical applicability of PCGSR. (By contrast, actual computational chemistry methods such as DFT are applicable to materials/molecules in general.) Although the paper does show results of transferability to a copper lattice compressed volumetrically by 50%, this compressed lattice is not something that can physically exist in the real world and is thus not of practical interest either. An experiment that would better showcase PCGSR would be to train on data from a non-trivial class of materials/molecules to discover 1 formula that accurately describes all of them.
4. Another main selling point of the paper is that it addresses the issue where previous methods do not search efficiently because of semantically equivalent expression trees. However, this issue has already been addressed in the literature in many ways—see the brief review given in the introduction section of Huynh et al. 2016 [2].
5. Another main selling point of the paper is its use of neural-guided search (deep RL + MCTS). This idea is also not new. See, e.g., Landajuela et al. 2021 [3].
6. The results in Table 1 are not particularly strong given that PCGSR does not outperform SPL or NGGP with great statistical significance ($1.3\sigma$ and $1.0\sigma$, respectively).

[1] Charles Fox, Neil D Tran, F Nikki Nacion, Samiha Sharlin and Tyler R Josephson, “Incorporating background knowledge in symbolic regression using a computer algebra system.” *Mach. Learn.: Sci. Technol. 5*, 2024.

[2] Quang Nhat Huynh, Hemant Kumar Singh and Tapabrata Ray, “Improving Symbolic Regression through a Semantics-driven Framework.” *IEEE SSCI*, 2016.

[3] Mikel Landajuela, Brenden K. Petersen, Sookyung Kim, Claudio P. Santiago, et al., "Discovering symbolic policies with deep reinforcement learning." *ICML*, 2021.

**Questions:**

1. The introduction of deep RL with the SGNN feels quite costly to me. What do the tradeoffs look like for the accuracy gains from the SGNN vs. the extra computational costs incurred? Or are these costs balanced out by the gains from the extra guidance from the SGNN in the later parts of the training? Table 2 shows # of evaluations as a cost metric, but that doesn’t take into account the forward and backward passes of the SGNN, so I wonder what the wall-clock time looks like with vs. without the SGNN?
2. The paper frequently mentions that the SGNN encodes physics constraints, but what people typically think of as “physics constraints” are a lot more involved than the ones considered. For example, I think of time/spatial symmetry and conservation of energy/momentum etc. Do you have a systematic way to incorporate constraints likes these in PCGSR?
3. It would be nice to contextualize the results in Table 3. What is the typical MAE in DFT calculations? What is typically the accuracy needed for an energy calculation to be practically useful?
4. What do the costs look like in Table 1 (e.g. # evaluations, wall-clock time)? Although the max # evaluations (500k) and the time limit (24hrs) were the same for all methods compared, it could be that, on easy tasks, PCGSR takes longer than the baselines. It would be helpful to see a breakdown over cost, e.g., the recovery rate when the time limit is 2, 4, 6, 12, 24 hours.

Corrections:
1. Abstract line 1: “explainable” -> “explainability”
2. Line 118: $f \subset Q$ – I don’t think “$f$ is a subset of $Q$” is the intended meaning here?
3. Lines 230 & 239: “We” -> “we”. Line 321: “we” -> “We”
4. Lines 421–424 (about Pauli’s exclusion principle): There’s a misunderstanding of Pauli’s exclusion principle here. The phenomenon described in lines 421–424 is just electrostatic repulsion (Coulomb’s law) and has nothing to do with Pauli’s exclusion principle. Pauli’s exclusion principle states that two fermions cannot be in the same quantum state. As an example, electrons in an atom cannot occupy the same quantum state; as a result, because each electron has two spin states, every atomic orbital can hold at most two electrons, resulting in the structure that we see in the periodic table (2 elements in row 1, 8 elements in row 2, etc.).

Suggestions:
1. Given my criticisms of the paper, the following reworked story would be more convincing to me: "We show that using a graph neural network to predict next actions in neural-guided MCTS outperforms previous approaches to neural-guided search for SR." That's it--nothing about the symbolic graph (Weakness 4) or physics constraints (Weaknesses 1-3).

---

> ### Author Response · Authors · 2024-11-22
> **Rebuttal to the reviewer QFds [1/4]**
>
> We appreciate the reviewer QFds' detailed comments and critiques. To help address the raised concerns about the physics constraints, related work, and experiments, please allow us to provide point-by-point responses to the comments, which will hopefully help clarify our model formulation and solution to avoid possible misunderstandings from the reviewer.
>
> **1) Physics Constraints [Responses to Weakness 1 and Question 2]**
>
> We summarize our responses to **Weakness 1** in **Global Rebuttal**. To address other concerns in **Weakness 1**, we clarify that our work does not aim to introduce *new physics constraints* for specific problems. Instead, we propose a general framework to tackle the reward sparsity issue arising from existing constraints. This self-learning framework leverages both post-constraint knowledge and MCTS's inductive bias.
>
> - **Tables 1 and 2**: These are designed to evaluate SGNN's effectiveness in promoting promising candidates in the search space due to its inductive bias and invariant graph representation. They provide a fair comparison against baselines without physics constraints, not to demonstrate constraint incorporation.
> - **Section 5 Example**: The comparison between non-SGNN-guided MCTS and PCGSR highlights SGNN's contribution in leveraging post-constraints.
> - **Section 3.4**: Constraints listed here are described as *general SR constraints* (line 293), not specifically as physics constraints.
>
> For **Question 2**, PCGSR is not intended to exhaustively demonstrate all possible constraints in the paper. Beyond the physics constraints in Section 5, additional examples, such as those mentioned by the reviewer, can also be incorporated following **Global Rebuttal Section 1** (*Systematic Implementation*). These constraints benefit similarly from reduced reward sparsity, as explained in **Global Rebuttal Section 2** (*Rationale Behind SGNN*).
>
> ---
>
> **2) Related Work on Physics Constraints** **[Responses to Weakness 2]**
>
> We summarize our response to **Weakness 2** in **Global Rebuttal Section 3** (*Comparison with Related Work*)
>
> ---
>
> **3) Related Work on Expression Equivalency** **[Responses to Weakness 4]**
>
> We thank the reviewer for suggesting the reference by Huynh et al. 2016 on expression equivalency. However, we respectfully disagree with the claim in Weakness 4 that “the expression equivalency problem has already been solved in many ways.”
>
> - After reading the reference, it is not clear how the paper's review is related to symmetries and invariances problems considered in our paper. The referenced work only mentions detecting identical terms via output sampling and pairwise comparison in GP for cross-over and mutation in the methodology section. However, it is significantly different from our work, as no symmetries or invariances are incorporated into their semantic representations.
> - By contrast, **PCGSR** introduces a symmetric invariant graph representation for expressions, leveraging equivalency without requiring output sampling.
> - Compared with other existing work mentioned in our introduction, our graph-based representation uniquely encodes symmetries and invariances with greater simplicity and accuracy, providing a novel solution to this problem.
>
> ---
>
> **4) Related Work on Neural-Guided Search** **[Responses to Weakness 5]**
>
> We appreciate the reference by Landajuela et al. 2021 but respectfully disagree with the assertion in Weakness 5.
>
> - The referenced work is a purely DeepRL-based framework without incorporating MCTS.
> - PCGSR addresses distinct challenges, including **expression invariant representation** and **physics constraint incorporation**, which differ from existing neural-guided search methods.
> - Our comparisons with NGGP (DeepRL + GP) and SPL (MCTS) in **Table 1** (in the main text of our paper) and **Table 4** (in the appendix) demonstrate PCGSR’s novelty and efficiency.
>
> Thus, we assert that PCGSR offers a novel and more efficient neural-guided search framework.

---

> ### Author Response · Authors · 2024-11-22
> **Rebuttal to the reviewer QFds [2/4]**
>
> **5) Copper Dataset Application**  **[Responses to Weakness 3 and Question 3]**
>
> **5.1) Practical Usage**
> Regarding Weakness 3, we respectfully disagree with the comment that the formula from the 32-atom copper dataset lacks practical utility:
> 1. **Framework Generality**: Our method is not limited by system size or atom types. The 32-atom copper dataset is chosen to serve as a benchmark simply because we want to compare PCGSR with prior work and demonstrate its advantages. The derived potential energy function can be applied to larger systems.
> 2. **Dataset Relevance**: The 32-atom copper dataset is representative of materials studies, consistent with the average system size (~31 atoms/structure) of the widely used MPtraj dataset (Deng et al., 2023 [1]).
> 3. **Scalability**: While DFT methods are limited to simulating small systems (1–1000 atoms) over very short timescales (a few picoseconds), our method enables large-scale simulations (millions of atoms) over extended timescales (microseconds)- which can then help determine many physical properties of materials that are not possible for quantum chemistry methods such as DFT. The analytical function from PCGSR can be applied to study real material problems with sizes far beyond 32 atoms for long-time dynamics that are completely inaccessible to DFT or other quantum chemistry methods -- which is the key purpose of developing force field based on accurate analytical potential energy functions from the PCGSR method. As a result, one can simulate the mechanical deformation process of single or polycrystalline copper consisting of millions of copper atoms using large-scale molecular dynamics simulations with the potential energy function developed here. For example, studying the novel stacking of copper in the incubation period of crystallization (Liu et al., 2023 [2]) requires the simulation of millions of copper atoms.
> 4. **DFT-Level Accuracy**: The dataset is derived from DFT calculations, enabling DFT-level accuracy but without explicit electron degree of freedom - that's the whole purpose of developing potential energy function (or, machine learning force field) from quantum chemistry datasets such as DFT. The high accuracy, the analytical nature, and the proper limiting trend at short bond length make this method and the derived potential energy function particularly useful.
>
> **5.2) Copper Dataset with Compressed Volume**
> We also respectfully disagree with the comment in Weakness 3 that "a copper lattice compressed by 50\% volumetrically is unrealistic":
> - **Scientific Context**: Understanding matter in extreme conditions such as high pressure and high temperature is an important and active subject of materials research. Copper is one of the systems of particular interest. As shown in McCoy et al., 2017 [3], the density in experiments reaches ~18 g/cm$^3$, double the density at the standard condition of 8.95 g/cm$^3$, that is, the volume contraction by 50\%. Another example is done by Fratanduono et al., 2020 [4] at the National Ignition Facility (NIF) at the U.S. Lawrence Livermore National Laboratory (LLNL), in which the solid copper was even compressed to ~28 g/cm$^3$ at terapascal conditions, corresponding to ~67\% volume contraction.
> - **Practical Necessity**: Beyond scientific motivation, another key motivation to apply large compression is to provide a more accurate trend away from equilibrium towards $r \rightarrow 0$. This is particularly important as the machine learning interatomic potentials or machine learning force fields very often have wrong limiting behavior. When they were applied to simulating long-time dynamics at high temperature or high pressure, there will be an increasing probability of ``direct crossing or fusion'' of atoms which are pure artifacts due to the wrong limiting trend, consequently, the results can be completely nonphysical and wrong.
>
> **5.3) Typical MAE**
> In response to Question 3:
> - DFT achieves MAE of 10–20 meV/atom for energy predictions.
> - Bond lengths and lattice parameters are typically accurate to within 1–2\% of experimental values.
>
> ---
>
> **References**
> 1. Deng, Bowen, et al. "CHGNet as a pretrained universal neural network potential for charge-informed atomistic modelling." Nature Machine Intelligence 5.9 (2023): 1031-1041.
> 2. Liu, Songling, et al. *Tetrahedral Stacking of Copper in Its Incubation Period of Crystallization.* *J. Phys. Chem. C* 127.44 (2023): 21816–21821.
> 3. McCoy, Chad A., et al. *Absolute Measurement of the Hugoniot and Sound Velocity of Liquid Copper at Multimegabar Pressures.* *Phys. Rev. B* 96.17 (2017): 174109.
> 4. Fratanduono, D. E., et al. *Probing the Solid Phase of Noble Metal Copper at Terapascal Conditions.* *Phys. Rev. Lett.* 124 (2020): 015701.

---

> ### Author Response · Authors · 2024-11-22
> **Rebuttal to the reviewer QFds [3/4]**
>
> **6) Computational Costs of PCGSR** **[Responses to Questions 1 and 4]**
>
> **6.1) Performance-Cost Trade-Off**
> PCGSR achieves a superior balance between performance and computational cost compared to existing methods:
> - **Deep RL-based Methods (e.g., DSR)**: Strong inductive bias but high computational cost and limited exploration.
> - **Naive MCTS Methods (e.g., SPL)**: Low cost and strong exploration but unable to incorporate constraints and inductive bias during random simulations.
> - **PCGSR's Advantage**: SGNN-guided MCTS combines the strengths of DSR and SPL, achieving costs comparable to NGGP (Neural Guided GP: DeepRL + GP) with better performance shown in our reported results in Table 1 and Table 4 of the paper.
>
> **6.2) Adaptive Computational Cost**
> PCGSR provides flexible cost control through the $\epsilon$ coefficient (line 258, Appendix B.4):
> - **Light Cost**: Set $\epsilon = 1$ to disable SGNN, matching SPL's cost for simpler problems.
> - **Full Capability**: Set $\epsilon = 0$ to leverage SGNN fully for complex problems, matching DSR’s cost.
> - **Dynamic Adjustments**: $\epsilon$ can adapt during training, enabling cost control based on problem complexity.
>
> **6.3) Training Time and Efficiency**
> The following presents comparisons of recovery rates, evaluations, and training times for Table 2 in the paper:
>
> | Model                | PCGSR  | MCTS-1  | MCTS-2  | MCTS-3  |
> |-----------------------|--------|---------|---------|---------|
> | **Recovery Rate (\%)** | 82.5   | 83.75   | 77.5    | 75.0    |
> | **Evaluations**       | 96,221 | 121,853 | 237,923 | 285,284 |
> | **Training Time (s)** | 1847.4 | 2485.8  | 4306.4  | 5848.3  |
>
> - **Efficiency Comparison** (evaluations/second):
>   - PCGSR: 52.08, MCTS-1: 49.02, MCTS-2: 55.25, MCTS-3: 48.78.
>   - SGNN improves efficiency by reducing the number of evaluations required and bypassing costly coefficient fitting for simulations guided by SGNN.
>
> For Training Time in Table 1: PCGSR -- 4.3 hours, SPL -- 5.2 hours, NGGP -- 5.7 hours, AI Feynman 2.0 -- 7.7 hours, and GP -- 3.8 hours
>
> **6.4) Key Insight**
> The most computationally expensive part is coefficient fitting for constants, rather than SGNN's forward/backward passes. SGNN reduces evaluations, making PCGSR more efficient overall by minimizing costly constants fitting and leveraging forward predictions for efficiency.
>
> ---
>
> **7) Experimental Results** **[Responses to Weakness 6]**
>
> Our conclusions are supported by comprehensive benchmarks across diverse datasets using multiple evaluation metrics:
> - **Table 1**: PCGSR achieves the lowest average recovery rate while maintaining the lowest complexity, highlighting its predictive capabilities in SR.
> - **Table 2 (Ablation Studies)**: Validates the efficacy of SG and SGNN, reflected in improved recovery rates and reduced evaluations.
> - **Table 4 (Appendix)**: Shows PCGSR's superior recovery rate over SPL and NGGP, particularly for complex tasks including the ones in Nguyen-12.

---

> ### Author Response · Authors · 2024-11-22
> **Rebuttal to the reviewer QFds [4/4]**
>
> **8) Notations** **[Responses to Corrections 1–4]**
>
> We thank the reviewer for these suggestions and will update the manuscript accordingly:
> 1. In **Abstract line 11**, we will retain “explainable” rather than “explainability,” as it better aligns with its role as an attributive modifying “machine learning (ML) methods.”
> 2. In **line 118**, we will revise to $f = [\phi_i\|\phi_i \in Q]$.
> 3. We will revise the presentation as suggested.
> 4. **Pauli's Exclusion Principle:** Regarding Correction 4, we partially disagree with the suggestion that the interaction at $r \rightarrow 0$ is purely electrostatic (Coulomb’s law) and unrelated to Pauli’s exclusion principle:
> The Pauli's exclusion principle is not only applied to the case of individual isolated atoms (which then determines how the periodic table is organized as the reviewer mentioned), but also are strictly followed even when atoms get closer and form bonds in molecules and solids. It's a universal principle. It is true that when $r \rightarrow 0$, the dominating interaction is nucleus-nucleus electrostatic interaction, but this is a subtle consequence of Pauli’s exclusion principle - which is hard to elaborate in the main text due to the limited space. Let's imagine two hydrogen atoms H$_A$ and H$_B$ getting closer and closer, each carrying one proton and one electron. If it was pure electrostatic repulsion, then two protons and two electrons should give rise to a total of four pairs of electrostatic interaction -- two repulsive ones from proton-proton and electron-electron electrostatic interaction and two attractive ones from the two pairs of proton-electron interaction. If so, they should just simply cancel each other with zero total energy at any distance, since the charges carried by electron and proton are exactly opposite with the same amplitude. Subsequently, there shouldn't be an equilibrium bond length for hydrogen H$_2$ molecule of 0.74 $\overset{\circ}{A}$, as the total energy would be the same for any separation distance.  In fact, the electron is fermion where Pauli's exclusion principle has to be considered. When atoms get closer to each other, the wavefunction of electrons surrounding the nucleus starts to have significant overlap with each other, and electrons would occupy the states that are already filled by other electrons. However, due to the Pauli's exclusion principle, no more than one electron can occupy the same state with the same four quantum numbers ($n$, $l$, $m_l$, $m_s$), hence the orbitals from individual atoms have to hybridize and form molecular spin-orbitals with unique quantum numbers, resulting in bonding and antibonding states available for electrons to occupy. When the nuclei get very close to each other, the electrons especially those in the antibonding orbitals will be pushed away from the nuclei, leaving the nuclei largely unscreened. Consequently, the nucleus-nucleus electrostatic interaction dominates at a very short distance. So, it is true that when $r \rightarrow 0$, the dominating interaction is nucleus-nucleus electrostatic interaction as the reviewer mentioned, but this is a subtle consequence of the Pauli’s exclusion principle as elaborated above. To address the reviewer's suggestion, we will include more detailed explanation on this to the revised manuscript.
>
> ---
>
> Overall, we thank the reviewer again for the detailed comments. We have already shown the Graph Neural Network (GNN)-based PCGSR outperforms several types of SR baselines through comprehensive benchmarking on different datasets with the recovery rate, evaluation numbers, and physical significance. We believe that it is the rationale behind the GNN (representations for invariance and encoding for priors of constraints and inductive bias) that makes our work effective and unique.
>
> Sincerely,
>
> The Authors

---

> ### Comment · Reviewer_QFds · 2024-11-22
> **Response to authors 4/4**
>
> 1. I agree with the authors--I had misparsed the sentence, my apologies.
> 2. If this represents a list of elements from $Q$, then that makes sense to me.
> 3. Cool!
> 4. Thanks for the clarification! I agree with the authors that a proper quantum treatment inevitably involves the Pauli exclusion principle since the system has multiple electrons, and that $E \to \infty$ as $r \to 0$ is dominated by the nucleus-nucleus electrostatic interaction due to less electron screening. I have one clarification question:
> - "When the nuclei get very close to each other, the electrons especially those in the antibonding orbitals will be pushed away from the nuclei, leaving the nuclei largely unscreened." Do I understand correctly that you're referring to the non-bonding electrons (ones that are in inner shells of the atoms), whose corresponding antibonding molecular orbitals are filled?
>
> Although I agree with the authors' explanation of the subtle connection between the Pauli exclusion principle and reduced electron screening, a few physics/chemistry people that I talked to agree that, if one were to give a short three-word description of the phenomenon that $E \to \infty$ as $r \to 0$, one should mention the electrostatic repulsion between the nuclei as that is the dominating effect. Yes, the reduced electron screening allows for this effect to be not canceled out, but I would suggest putting this explanation in the appendix, and replace every instance of "Pauli exclusion principle" with "electrostatic repulsion" to point out the more direct underlying phenomenon.
>
> Also, I would like to point out that a purely classical model of the electrostatic interaction *can* in fact account for reduced electron screening and thus the phenomenon that $E \to \infty$ as $r \to 0$. Modeling the nuclei in a molecule as point charges and electrons as charge densities, the total electrostatic potential energy has 3 components:
> * Internucleic energy. This energy goes to $+\infty$ if the distance between any two nuclei goes to zero.
> * For each nucleus, its energy with the electron charge density. This energy is always finite since the electron charge density has no delta functions.
> * The electron cloud's energy with itself. This is also finite because it has no delta functions.
> Summing the three components together, we get the behavior that the total energy diverges when the distance between two nuclei goes to zero.
>
> Here, "reduced electron screening" exhibits itself as the fact that the attractive energy between a nucleus and the electron cloud of another nucleus does not shoot to $-\infty$ as the nuclei approach each other, instead approaching a constant due to the fact that the electron density is distributed across space.

---

> ### Comment · Reviewer_QFds · 2024-11-22
> **Response to authors 1/4**
>
> I appreciate the authors' detailed response, and I mostly agree with what the authors said.
>
> I'm going to try to rephrase what you said in your Global Rebuttal to make sure I understand correctly (please correct me if I'm misunderstanding):
>
> * Non-deep-learning approaches to SR do not efficiently/flexibly incorporate complex constraints because the expression generation component is fixed, so a large number of generated expressions are wasted.
> * We hypothesize that, by having a deep NN that generates expressions, it can learn through RL to generate expressions that satisfy these constraints, thus being more efficient. We demonstrate this using an SGNN generative model over expressions that learns through RL to generate expressions that satisfy constraints.
> * We find that, with pure RL (no search), such an approach can suffer from the sparse reward problem since the penalty is given only after a complete expression is generated, and constraint-satisfying expressions can be very sparse. We thus incorporate MCTS to mitigate this sparse-reward problem.
>
> Do I understand correctly? If so, I have reservations regarding the amount of experimental evidence for these claims, which I have summarized in my [Overall summary](https://openreview.net/forum?id=Ia17iAtr0P&noteId=7iYcuel0kl) comment below.

---

> ### Comment · Reviewer_QFds · 2024-11-23
> **Response to authors 2/4**
>
> **5.1)** I will try to rephrase to make sure I understand correctly. My concern was that, to apply your approach to a new material, you will have to first get MD simulations of that material using DFT, and then apply SR to get a formula that would only work for the same system, so why bother with the SR? However, you're saying that the actual use of SR is:
> 1. For a new material, take a small set of atoms (e.g. 32 atoms), and apply DFT simulations to get data.
> 2. Apply SR to get an energy formula *specific to that material*.
> 3. Although the formula is specific to that material, it can be scaled up to a large system (e.g. 1000 atoms) where DFT simulations would become intractable. And the force field derived from the formula can be used for efficient MD simulations of these large systems as well. In summary, the formula is learnt from small DFT simulations and then used for large systems of the same material.
>
> If this is the right understanding, I would suggest adding this scientific motivation to the main text.
>
> **5.2)** Thanks for the clarification on the broader scientific context. It would be nice to add this to the paper.
>
> **5.3)** Thanks for your answer. I would recommend adding something like this to the paper for better contextualization for the reader.
>
> **Additional question:** What's the evidence that your formula generalizes to larger systems (e.g. thousands or millions of atoms), which is the point of having analytic expressions for energy if I'm understanding correctly?

---

> ### Comment · Reviewer_QFds · 2024-11-23
> **Response to authors 3/4**
>
> Thank you for your answers to my questions!
>
> I have **two addtional questions**:
> 1. If I'm understanding correctly, NGGP also addresses the tradeoff mentioned in 6.1. Is the improvement over NGGP then mainly due to the symbolic graph representation that minimizes redundancies?
> 2. Is there a comparison with NGGP in terms of computational costs? If I understand correctly, I would expect PCGSR to be less costly since the SG representation allows for more efficient exploration and search, reducing the number of evaluations. Is that understanding correct?

---

> ### Comment · Reviewer_QFds · 2024-11-23
> **Overall summary of my thoughts after reading author responses**
>
> I thank the authors for their detailed responses and explanations. They have cleared up some misunderstandings I had about the paper--I recommend that the authors greatly revise their paper (especially the abstract) to reduce such misunderstandings if the paper is accepted.
>
> I have revised my "Contribution" score from 1 to 2, but have kept my "Soundness" score at 1 and overall score at 3 (reject).
>
> After reading the author responses, the motivation for the paper has now become a lot clearer, and I can see a version of the paper that's worthy of publication. However, according to my revised understanding of the paper's claims, more experiments are needed to support them.
>
> The paper tries to do two things: (1) show that the symbolic graph representation allows for more efficient search over expressions; (2) show that deep-RL + MCTS improves SR in domains that incorporate "post-constraints". I no longer have objections against (1) thanks to the authors' responses (I recommend highlighting the cost effectiveness more in the paper), but regarding (2) two main reservations remain:
> * Limited evaluation. Only 1 domain (energy in materials) is presented, with only one example of a post-constraint ($f(r) \to \infty$ as $r \to 0$). The paper would benefit from more examples of post-constraints to show that deep-RL + MCTS handles them better than pure deep-RL or pure MCTS/GP approaches.
> * Limited comparison. To properly show deep-RL+MCTS better incorporates post-constraints, the experiments table should compare across three groups: deep-RL + search/GP methods (including PCGSR), deep-RL-only methods (such as DSR), and search/GP-only methods (such as the ones already in the table). The current table omits deep-RL-only methods entirely.
>
> Apologies for not bringing these two points up earlier, as it was not clear to me that the paper had intended to claim (2).
>
> If these two points are addressed by the end of the discussion period, I will be happy to increase my score depending on how well they are addressed.

---

> ### Author Response · Authors · 2024-11-25
> **Response to Reviewer QFds's Reply**
>
> **9) Response to Reviewer QFds's Reply 1/4**
>
> - We sincerely appreciate the reviewer's efforts. We agree with the reviewer's first and second understanding of our work, but would like to note the difference from the reviewer's third statement:
>
>   - The pure Deep-RL method (e.g. DSR) does suffer from the sparse reward problem, but due to different reasons from the first understanding that the search-based methods (naive MCTS and GP) suffer from.
>
>   - Pure Deep-RL methods do not have such a *"fixed generation component"*, and deep learning neural networks enable learning post-constraints during sampling. The problem with pure Deep-RL lies in that neural networks being used for every sampling makes it costly, and many of these methods lack sufficient exploration, making them difficult to learn from the sparse rewards and easy to be trapped in the local optima. We thus incorporate SGNN-guided MCTS to balance random sampling (exploration) and SGNN prediction (exploitation).
>
> ---
>
> **10) Response to Reviewer QFds's Reply 3/4**
>
> 1. Regarding the first question, yes, that's the main reason. For the other reason, NGGP uses RNN to generate populations for GP, which combines Deep RL with search to stabilize the neural network training for sparse reward problems. But in the search component, they still have the *"fixed generation component"* (random cross-over and mutation) that cannot incorporate the post-constraint knowledge, making it less effective as our PCGSR has done.
>
> 2. Concerning the cost comparison, we have provided the training time of PCGSR and NGGP of Table 1 in the **Rebuttal Section 6.3** of the original rebuttal. Yes, what the reviewer wrote is indeed one of the reasons why PCGSR is less costly. The second reason is what we have explained in the first point of this section, *"fixed generation component"*. The third reason is that NGGP needs to do costly constant fitting for every search, but SGNN can encode constant fitting knowledge by reward prediction and can skip the constant fitting.
> ---
>
> **11) Response to Reviewer QFds's Reply 2/4**
>
> 1. Regarding the reviewer's understanding and suggestion on the actual use of SR, the answer is yes, and we thank the reviewer for the thoughtful suggestion. Indeed, as mentioned by the reviewer, the key advantage is that, by learning the analytic formula with PCGSR from a small but accurate quantum chemistry calculated dataset, one can use it for large systems of the same materials. We also want to emphasize that this approach can be extended to a full periodic table and develop analytic potential energy functions for a foundation model using the large MPtraj dataset etc. It is beyond the current scope, but we will explore this direction in the future.
>
> 2. Regarding the additional question on generalization to larger systems, it requires an additional implementation and benchmark of a separate module in a large-scale molecular dynamics package such as (LAMMPS), which is beyond the current scope of the work. We plan to implement it and conduct additional calculations in the future.
>
> 3. We appreciate the suggestions from the reviewer. We will include more details about the scientific motivation and the broader scientific context in the main text, and revise the manuscript accordingly. We will also include additional background information such as the accuracy of typical DFT methods for better contextualization.
>
> ---
>
> **12) Response to Reviewer QFds's Reply 4/4**
>
> 1. The understanding of the second point is correct. We agree with the reviewer that the electrostatic repulsion between nuclei becomes the dominating effect at very short bond lengths. We also agree with the reviewer to include a more explicit explanation in the appendix and will upload a revised manuscript.
>
> 2. Regarding the clarification question about whether we are *“referring to the non-bonding electrons (ones that are in inner shells of the atoms), whose corresponding antibonding molecular orbitals are filled?”*, the answer is yes.  At short distances, the non-bonding electrons in inner shells (also called core electrons) start to hybridize and form molecular orbitals. Some of the molecular orbitals will exhibit bonding-like shapes and others exhibit anti-bonding-like shapes. In terms of electron density distribution, those anti-bonding orbitals will be pushed further away from the bonds and the nuclei, further reducing the screening around the nuclei.
>
> 3. Regarding the reviewer's comment that *“a purely classical model of the electrostatic interaction can in fact account for reduced electron screening and thus the phenomenon that $E \rightarrow \infty$ as $r \rightarrow 0$”*, we agree with the reviewer that the proposed classical treatment can indeed lead to a similar conclusion for the limiting trend since the electron density is a probability distribution and thus the repulsive or attractive energy involving electrons will not have $\delta$-function in their expression.

---

> > ### Comment · Reviewer_QFds · 2024-11-25
> > **Response to authors (9-12)**
> >
> > **9)** Thanks for the additional explanation--it confirms my understanding.
> >
> > **10)**
> > 1. Since the results in Table 1 are on benchmarks that do not involve post-constraints, I assume the other reason you're mentioning would apply only to Section 5?
> > 2. Thanks for pointing out the compared computational costs--apologies for missing it in your original rebuttal. I couldn't find the third reason you mentioned in the paper. It could be nice to have an elaboration on that in the paper, perhaps in a separate analysis section that does the ablation study and details the various features of PCGSR and how they contribute to its efficacy. The ablation study would also include ablating $R(s)$ to illustrate your point.
> >
> > Also, when you say "reward" $R(s)$ of a state, I suppose you really mean the "value function" $V(s)$?
> >
> > **11)**
> > 1. Thanks for the confirmation. The future sounds exciting!
> > 2. I see. I would like to see some evidence that the SR formula generalizes well to larger systems--is there something that you can cite to support this statement?
> >
> > **12)**
> > We seem to be in agreement about this issue.

---

> > > ### Author Response · Authors · 2024-12-01
> > > **Responses to Reviewer QFds's Reply 9-12**
> > >
> > > **Response to Reviewer's Reply 10**
> > >
> > > We thank the reviewer for their additional questions and patience in awaiting our responses. Below, we provide detailed explanations:
> > >
> > > 1. **Improvement from Capturing Hidden Post-Constraints**
> > >    The improvement due to capturing hidden post-constraints also applies to Table 1, as explained in **Rebuttal 13**.
> > >
> > > 2. **Explanation of Predicted Prior and Predicted Rewards**
> > >    We addressed the third reason in **Equation 2** and **Lines 257–261** of the paper, which detail how SGNN predictions on prior probabilities and rewards replace the random rollout (*"fixed generation component"*) of MCTS. The random rollout requires constant fitting, as explained in **Appendix B.1**.
> > >    - In the revised manuscript, we have elaborated on:
> > >      - The rationale of the **predicted prior** in the *"Constraints Incorporation"* section.
> > >      - The rationale of the **predicted rewards** in the *"Cost-Effectiveness"* section.
> > >    - We note that Table 2 already includes predicted rewards $R(s)$ by SGNN in PCGSR and MCTS-2 (MCTS-GNN), because the predicted prior $\textbf{P}(s)$ and predicted rewards $R(s)$ overall are integrated with SGNN to replace the random rollout in MCTS.
> > >
> > > - Yes, $R(s)$ serves as the value function.
> > >
> > > ---
> > >
> > > **Response to Reviewer's Reply 11**
> > >
> > > As one of such examples, the work by Cherukara et al., 2016 studies a bond order potential (BOP) to accurately describe the energetics, thermal, and mechanical properties of stanene. This model is trained on molecular dynamics DFT simulations of \~ 10 atoms system with 1000 steps to investigate the temperature-dependent thermal conductivity of stanene nanostructures and then generalizes to a stanene supercell (67,716-atom)  by an average of over 1 million steps.
> > >
> > > **Reference**
> > >
> > > Cherukara, Mathew J., et al. "Ab initio-based bond order potential to investigate low thermal conductivity of stanene nanostructures." The journal of physical chemistry letters 7.19 (2016): 3752-3759.

---

> > > > ### Comment · Reviewer_QFds · 2024-12-03
> > > >
> > > > ### 10
> > > >
> > > > 1. Got it--thanks!
> > > >
> > > > 2. Having a section explaining the rationale of the value function and how it reduces costs is appreciated. I will note that the performance gains here are fundamentally due to RL and not a unique contribution of PCGSR: most approaches to RL involve jointly learning a policy and value function that converge towards the optimal policy and the correct value function. So I would recommend framing your discussion as a benefit of RL in general in your paper.
> > > >
> > > > Because the standard terminology for your $R(s)$ is "value function" (usually denoted $V(s)$), I would recommend replacing instances of "reward" and "$R(s)$" in the paper with "value" and "$V(s)$", respectively. (An exception is lines 228-229, where "reward" matches the standard definition of "reward".)
> > > >
> > > > ### 11
> > > >
> > > > I have read the introduction of the referenced paper and agree that it's a good reference showing the practical utility of interatomic potentials for MD simulations at large scales. I would recommend including the reference in the manuscript so that the reader can understand the practical significance of applying SR to the discovery of interatomic potentials.

---

> ### Author Response · Authors · 2024-11-25
> **Response to Reviewer QFds's Overall Reply**
>
> **13) Response to Reviewer QFds's Overall Reply**
>
> We are glad that our responses have helped clarify the reviewer's questions/misunderstanding and are truly thankful for all the constructive suggestions to strengthen our paper with better presentation quality. For the remaining concerns, we would like to provide our explanations in the following:
>
> - **SGNN Universal Encoding** Regarding the first remaining concern, We want to clarify that post-constraints not only encompass the physics post-constraints but also include hidden post-constraints naturally coming from the SR problems (e.g. $\log(A+?)$ will not be valid in the real-number realm if we further sample "$?$" to be "$f(B)$" but $A+f(B) < 0$).
>
>   - SGNN's encoding of post-constraints (physics constraints and hidden constraints) by predicted prior and encoding of constant fitting by predicted rewards are universally applicable in SR. The ablation study for the updated Table 2 in **Rebuttal Section 6.3** comprehensively demonstrates the universal capabilities of SGNN encoding to improve the search quality and efficiency as explained in that rebuttal and in **Continued Rebuttal to Reviewer 4ixR Section 3 Point 2**, along with the benchmarking results in Table 1 and Table 4 in the paper.
>   - Though SGNN-guided MCTS is comprehensively benchmarked as stated above, we show the example in the **Application** section in our paper to emphasize the problem complexity and reward sparsity for real-world cases and show our proposed PCGSR's superiority in performances and physical significance, compared with other SR baselines that mostly were only evaluated in synthetic datasets such as the Feynman dataset in SRBench.
>   - We agree with the reviewer that it would always be ideal if we could show more complicated physical post-constraints and more domain applications. However, due to the page limit, the lack of physically informed SR baselines in similar problems, and the limit of time during the rebuttal period, we would include them in our future research. We believe that the current experiments including real-world applications have consistently and reliably demonstrated the significance of our presented work to support our claims regarding PCGSR as well as its constituent components, including SGNN-guided MCTS.
>
> - **Added DSR Results** Regarding the second remaining concern, we agree with the reviewer that adding pure Deep-RL
>   methods to our benchmarks would provide deeper insight into why PCGSR outperforms existing SR methods. We will update Table 1 in the revised paper as the following table. The training time for DSR is 6.1 hours along with the run-time of other methods as reported in **Rebuttal Section 6.3**. In this experiment, DSR also has a lower recovery rate as explained in **Rebuttal Section 9**. For Table 3 of the **Application** section in the paper, DSR is not evaluated due to the lack of support for the basic scalar output constraint to generate basic values for evaluation.
>
> |         Model         |      PCGSR      |       NGGP       |        SPL       |        GP        |        DSR       |  AI Feynman 2.0 |
> |:---------------------:|:---------------:|:----------------:|:----------------:|:----------------:|:----------------:|:---------------:|
> | **Recovery  Rate (\%)** | 62.18$\pm$ 3.00 | 60.22 $\pm$ 2.27 | 58.93 $\pm$ 3.73 | 20.17 $\pm$ 3.21 | 23.62 $\pm$ 2.28 | 51.26$\pm$ 5.82 |
> | **Complexity** |      30.56      |       36.57      |       32.48      |       46.05      |       22.78      |      42.01      |
>
> - We will incorporate the following discussions, corrections as well as additional experimental results as suggested by the reviewer in the revised paper:
>   - Revised abstract for clear and accurate descriptions of our contributions.
>   - Clarifications for all the terms mentioned in **Rebuttal Section 8**, including *"electrostatic repulsion"* for the constraint.
>   - Improved descriptions in the **Introduction** section for more related work review regarding expression equivalency and physics constraints incorporation.
>   - Improved descriptions of the constraint incorporation and the rationale behind SGNN in the **Introduction** and **Constraints Incorporation** section in the paper based on the **Global Author Rebuttal**.
>   - Improved **Experiments** section to include the updated experimental results in **Rebuttal Sections 6 and 13** highlighting the cost-effectiveness of PCGSR.
>   - Improved **Application** section to include the physical insights discussed in **Rebuttal Section 5**.

---

> ### Comment · Reviewer_QFds · 2024-11-25
> **Response to authors (13)**
>
> **Regarding concern about limited post-constraints in evaluation:** Thanks for the clarification on hidden post-constraints. I have a **follow-up question**:
> * If the claim is that these hidden post-constraints would result in the reward sparsity problem that makes other models less performant, why is it the case in Table 2 that there's a significant drop from PCGSR to MCTS-2? It seems that a similar drop in Table 1 would put MCTS-2 below NGGP.
>
> **Regarding concern about limited comparison:** _**To clarify, I was referring to Table 3**_, but I appreciate adding the DSR result to Table 1 as well.
>
> **I also have an additional question:**
> * For the ablation study, I realized it was conducted on a size-8 subset of the Feynman dataset. How were they chosen? (I suspect they might be the easier ones given the numbers are higher than in Table 1.) Given the number of equations is only 8, how significant are the differences across the methods reported in the table?
>
> Regarding improvements made to the paper, there are two more that I strongly recommend:
> * I would recommend changing "physics constraints" to something else throughout the paper. Maybe just "constraints"?
> * Oftentimes the paper implies somehow your model can *enforce* physical constraints in the expression generation process (e.g., in the name "Physics-Constrained Graph Symbolic Regression" and in the phrase "incorporate physics constraints"). Instead, you're really trying to say the generative model can better learn to generate expressions that satisfy constraints.

---

> ### Comment · Reviewer_QFds · 2024-11-30
> **Follow-up**
>
> I thank the authors for their responses to my concerns and questions. I would appreciate it if they could respond to my [newest concerns and questions](https://openreview.net/forum?id=Ia17iAtr0P&noteId=5CBY8FLqOT).
>
> In the meantime, given the satisfactory responses to some of my original questions and concerns, I have revised my Soundness score to 2 and Overall score to 5 (weak reject).

---

> ### Author Response · Authors · 2024-12-01
> **Responses to Reviewer QFds's Reply 13**
>
> **Response to Reviewer's Reply 13**
>
> 1. **Concerns About Evaluation**
>
> - Because the dominant factor influencing the performances of the models in Table 1 and Table 2 is the redundant space rather than the reward sparsity, in the paper when comparing with PCGSR, MCTS-2 (MCTS-GNN) lacks the SG representation to compress the search space for efficient search, leading to the significant performance drop.
>
> - Though the SG representation is the main contribution to PCGSR for lower reward sparsity scenarios in Table 2, MCTS-2 (MCTS-GNN) without SG would still have advantages over NGGP through SGNN-guided MCTS as the following table shows:
>
>
> |         Model         |  PCGSR  |  MCTS-SG |  MCTS-GNN |  MCTS |  NGGP |
> |:---------------------:|:-------:|:-------:|:-------:|:-------:|:-------:|
> | Recovery  Rate (\%) |   82.5  |  83.75  |   77.5  |   75.0  |   75.0  |
> | Number of Evaluations |  96,221 | 121,853 | 237,923 | 285,284 | 261,943 |
>
> 2. **Table 3**
> - **DSR Cannot Generate Valid Expressions for Table 3**  DSR lacks support for the *"Scalar Output Constraint"*, making it unable to map the vector feature $r$ into the scalar output $E$ required for evaluation.
>
> - **Electrostatic Repulsion Post-Constraint Supports Table 3 Effectively**  The post-constraint $f(r \rightarrow \infty) \rightarrow \infty$ demonstrates its adequacy in handling Table 3 with the *Invalid Expression Comparison* as follows:
>   - In Table 3 (with the Electrostatic Repulsion constraint):
>      - **PCGSR** generates **65.7\%** invalid expressions.
>      - **MCTS-SG (MCTS-1)** generates **91.2\%** invalid expressions.
>   - In Table 2 (with only hidden constraints):
>      - **PCGSR** generates **12.3\%** invalid expressions.
>      - **MCTS-SG (MCTS-1)** generates **18.6\%** invalid expressions.
>
>   These results indicate that the $f(r \rightarrow \infty) \rightarrow \infty$ constraint provides adequate reward sparsity, serving as a realistic and effective example of real-world constraints.
>
> 3. **Additional Questions**
>
> - The benchmark problems in the Feynman dataset were chosen across different complexity, from easy to hard, to fairly compare efficiency and accuracy in different scenarios. This includes one expression of complexity 2, three expressions of complexity 3, one expression of complexity 4, one expression with complexity 7, one expression with complexity 9, and one expression with complexity 12.
>
> - To show the difference between Table 1 and Table 2, we further added NGGP in Table 2 in Section 1 of this rebuttal. Table 2 has a higher recovery rate because the original Feynman dataset for Table 1 has many problems with high complexity and dimensionality that are intractable for many of the competing methods given the standard running limit, leading to a diluted advantage of PCGSR. This is the reason for us to do the ablations in the tractable problems.
>
> 4. **Improvements**
>
> - We thank the reviewer for their valuable recommendations.
> - The term *"Physics-constrained"* in our title highlights PCGSR's superiority in handling physics constraints, particularly its ability to generate physically meaningful results under sparse reward scenarios.
> - However, as suggested, we have revised the manuscript to first emphasize the **generalizability** of PCGSR in encoding various constraints, and then underscore its significance in solving real-world problems. This update aligns with the reviewer's thoughtful feedback.

---

> > ### Comment · Reviewer_QFds · 2024-12-03
> >
> > 1. Got it. So Table 1 and Table 2 results mainly show the advantages of SG and not SGNN because the hidden post-constraints are not strong enough, while Table 3 justifies the SGNN through the post-constraint.
> >
> > 2. I see. However, I do believe that a pure RL (no search) baseline (e.g., from a modification of PCGSR) would be good to have to justify the MCTS approach.
> >
> > 3. Got it. I believe it would be good to have a more systematic/justified way of choosing the problems and explicitly stating it in the manuscript.
> >
> > 4. The revision looks good to me, although I shall maintain that a reader who sees "physics-constrained" would expect more evaluations with physics constraints (in addition to the one regarding electrostatic repulsion).

---

> ### Comment · Reviewer_QFds · 2024-12-03
> **Final summary**
>
> I greatly thank the authors for engaging in productive discussions about the paper and appreciate their efforts in addressing my questions and concerns about the paper.
>
> After the discussion, I have increased my score from the original 3 (reject) to 5 (weak reject). While some of my concerns have been addressed, I believe there remains room for improvement regarding my [main concerns](https://openreview.net/forum?id=Ia17iAtr0P&noteId=7iYcuel0kl):
>
> * Regarding my concern that the only evaluation for post-constraints is the electrostatic repulsion case study, the authors argued that the main experiments also have "hidden" post-constraints. However, they also admit that these "hidden" post-constraints do not result in an amount of reward sparsity significant enough to justify GNN-guided MCTS. As a result, effectively the paper's claim that their approach better incorporates post-constraints (which they frequently emphasize in the paper and is in the title of the paper) is effectively only supported by experiments in one domain with a single example of a post-constraint.
>
> * Regarding my concern about the lack of comparison with a pure-RL algorithm in Table 3, the authors argue that DSR has a technical limitation that prevents it from being applied. However, I believe that a modified PCGSR baseline that is pure RL would be sufficient to justify the MCTS approach. (Similarly, the ablation study in Table 2 can have a pure-RL baseline as well to better support the paper's main claim.)
>
> While my concerns prevent me from recommending acceptance for the current version of the paper, I think it's going in an exciting direction and believe in the merits of the approach. I find the application to finding interatomic potentials exciting, and look forward to seeing other examples of physics constraints that PCGSR can easily incorporate.
>
> Sincerely,
>
> Reviewer QFds

---

> > ### Author Response · Authors · 2024-12-04
> > **Responses to Reviewer QFds' Final Summary [2/2]**
> >
> > 2. **Regarding Pure Deep-RL Methods in Table 3**
> >
> > - Table 2 focuses on ablation studies based on our enhancements to naive MCTS methods. We did not include DSR in this table because it belongs to a different branch of SR approaches. Moreover, baseline models SPL (naive MCTS) and NGGP have already demonstrated significant improvements over DSR in their paper, which is consistent with the results in our updated Table 1.
> >
> > - To address the reviewer’s concern, we attempted an additional ablation study by implementing a pure policy-gradient-based Deep-RL version of PCGSR using only GNN. However, after over eight hours of training, this implementation failed to converge and was trapped in local optima, yielding no valid SR solutions that satisfied the constraints. This demonstrates that RL alone, without search mechanisms, is insufficient to achieve the performance reported by the complete PCGSR implementation. We will report the full results in our final version of the paper.
> >
> > ---
> >
> > **Summary**
> >
> > We thank the reviewer for their extended discussions and thoughtful feedback. While SGNN plays a critical role in efficient constrained searches, it is the integration of **SG** and **SGNN-guided MCTS** that enables PCGSR to effectively navigate a compressed search space. We would greatly appreciate it if the reviewer could consider both components when evaluating the contributions of our work.

---

> ### Author Response · Authors · 2024-12-04
> **Responses to Reviewer QFds' Final Summary [1/2]**
>
> We thank the reviewer for their invaluable extended questions and suggestions throughout the discussion phase. While we agree with most of the raised concerns, we would like to provide additional experiments and explanations to clarify any misunderstandings regarding the contributions and scope of our work in response to the reviewer’s latest comments.
>
> ---
>
> **Response to Reviewer's Official Comment 10**
>
> We respectfully disagree with the reviewer's assessment that the performance gains in Table 2 are purely attributable to RL. The improvements observed in Table 2 are a result of our carefully designed components **Symbolic Graph (SG)** and **neural-guided MCTS** which work in synergy within PCGSR to enhance RL methods. Both the invariance encoding and the neural-guided MCTS are novel contributions that address prior limitations in symbolic regression (SR) approaches. Together, they form a unified and effective framework tailored to SR problems.
>
> ---
>
> **Response to Reviewer's Final Summary**
>
> 1. **Regarding the Evaluation of Post-Constraints**
>
> - > **Claim:** "However, they also admit that these 'hidden' post-constraints do not result in an amount of reward sparsity significant enough to justify GNN-guided MCTS."
>
>   - While we appreciate the reviewer's efforts in interpreting our experiments, we have to clarify that **we did not agree with the above statement**.
>     - In our **Responses to Reviewer QFds's Reply 13 Section 2**, we stated that the *"electrostatic repulsion"* post-constraint provides adequate reward sparsity compared to the hidden constraints.
>     - In the paper’s **Section 4.2**, we noted that while SGNN encoding effectively reduces the search size, it does not significantly impact the recovery rate due to less sparse rewards.
>     - These statements, however, do not lead to the conclusion that SGNN is unjustified. On the contrary, our experiments demonstrate that SGNN encoding of post-constraints reduces the search space by **21\%** and **16.6\%** (Table 2) and significantly reduces the running time (Table 1). This is detailed in our **Response to Reviewer QFds's Overall Reply Section 1** and the revised manuscript.
>
> - **On Domain-Specific Applications:**
>   While we respect the reviewer’s interest in exploring extended domain-specific applications, we emphasize that the focus of our work is to provide a **general framework** for incorporating constraints and addressing reward sparsity, rather than exhaustively demonstrating all possible domain applications.
>     - Our current benchmarks follow the same problems in Hernandez et al. (2019) as detailed in the **Application Section**, ensuring fair comparisons on standard real-world problems. PCGSR outperforms baselines not only in accuracy but also in satisfying more constraints (the only physically meaningful solutions except the ablation MCTS-GNN model) under high-reward sparsity.
>     - The comprehensive comparisons between existing work and the ablation model MCTS-SG already **demonstrate PCGSR’s efficacy in incorporating customized constraints and addressing reward sparsity**.
>     - Though incorporating more physics constraints can introduce more physical insights, **the number of physics constraints doesn't represent the sparsity of the rewards**. Our proposed reward sparsity of the electrostatic repulsion constraint explained in  **Responses to Reviewer QFds's Reply 13 Section 2** already satisfies the high reward sparsity problem in our scope.
>     - While extending our work to more domain-specific tasks could provide additional physical insights, we believe that **focusing on a representative problem does not diminish the significance of our contributions**. We appreciate the reviewer's suggestions and would leverage domain-specific tasks including ODE/PDE problems in our future research.

---

### Author Response · Authors · 2024-11-22
**Global Author Rebuttal**

We thank the reviewers for their time and efforts to provide thoughtful feedback. Below, we address the common questions and concerns regarding the physics constraints, clarifying the rationale and significance of SGNN to incorporate them to provide a clearer understanding of our model formulation and solution.

The "physics-constrained" method in our work refers to the SGNN mechanism learning the encoding of priors for hand-crafted constraints, rather than introducing new physics-specific constraints. We clarify key aspects below:

**1) Systematic Implementation to incorporate different constraints**
We classify constraints into two categories:
- **Pre-constraints**: Simple constraints testable during token sampling, such as mathematical rules or general symbolic regression (SR) constraints (Section 3.4).
- **Post-constraints**: Complex constraints requiring a complete expression for evaluation (e.g., $f(r \rightarrow 0) = \infty$ in Section 5.2).

**Strategies for incorporating constraints**:
- **Pre-constraints**: During token sampling, we zero out probabilities of actions violating constraints, preventing invalid expressions.
- **Post-constraints**: After generating a complete expression, we penalize invalid outputs with zero rewards.

This flexible strategy allows users to define customized constraints and assign them based on these categories.

**2) Rationale Behind SGNN**
Post-constraints cannot be directly encoded as pre-constraints to prevent invalid expressions during sampling, due to the requirement of full expression for testing, leading to sparse rewards in search spaces with complex constraints (e.g., Section 5). This challenge particularly affects sampling methods including naive Monte-Carlo Tree Search (MCTS) or Genetic Programming (GP), which fail to learn from constraint violations for certain training horizons (random simulation in MCTS, and random crossover and mutation in GP).

To address this, we propose **SGNN-guided MCTS** (Section 3.3):
- SGNN is trained (Eq. 5) to predict priors (Eq. 2), making it possible to integrate post-constraint priors into pre-constraint strategies.
- This mechanism enables SGNN to:
  - Leverage post-constraint insights during the MCTS simulation token sampling phase.
  - Discover hidden constraints and promote promising candidates.
  - Alleviate the issues due to reward sparsity and improve search efficiency.

**3) Comparison with Related Work**
We appreciate the reviewer QFds’s reference to Fox et al. 2024. However, our focus differs:
- **Existing works**: Propose new constraints or methods to incorporate them via hand-crafted priors or penalty functions.
- **Our work**: Tackles reward sparsity caused by post-constraints, introducing SGNN to self-learned priors that leverage post-constraint knowledge during sampling, improving exploration efficiency.

By integrating SGNN-guided strategies, our approach significantly enhances search performance in complex constraint environments.

---

### Author Response · Authors · 2024-12-01
**Summary of Manuscript Revisions**

We extend our sincere gratitude to all reviewers for their constructive feedback, which has greatly helped us enhance the quality of our presentation and experiments. Based on the reviewers' questions, suggestions, and our responses, we have made the following revisions to the manuscript:

- **Revised Abstract and Introduction (Constraints Section)**
  - Replaced all references to *"physics constraints"* with *"constraints"* to emphasize the generalizability of SGNN's encoding.
  - Highlighted *"physics constraints"* as a special case relevant to sparse reward scenarios and real-world significance.
  - Updated the review of physics constraints incorporation for better accuracy.

- **Revised "Constraints Incorporation" Section**
  - Introduced the concepts of *Pre-constraints* and *Post-constraints* along with their respective incorporation strategies.
  - Elaborated on the rationale behind SGNN's role in encoding *Post-constraints*.
  - Listed *physics constraints* as specific examples of post-constraints.

- **Added "Cost Effectiveness" Section**
  - Provided a detailed explanation of SGNN's role in reducing computational costs.
  - Discussed SGNN balancing the trade-off between exploration and exploitation compared to prior methods.

- **Updated "Experimental" Section (Table 1)**
  - Included DSR as an additional pure Deep-RL baseline.
  - Added running efficiency metrics for all models to provide a more comprehensive comparison across different SR methods.

- **Updated "Experimental" Section (Table 2)**
  - Renamed ablation models for improved clarity (e.g., *MCTS-1* → *MCTS-SG*, *MCTS-2* → *MCTS-GNN*, *MCTS-3* → *MCTS*). The missing Recovery Rate in Table 2 (previously noted in the Rebuttal to Reviewer QFDS[3/4]) will be recovered in our final version.
  - Included the running time for each model and added an efficiency analysis section to highlight PCGSR's cost-effectiveness.
  - Clarified the equation selection strategy in Appendix Section C.3.

- **Updated "Application" Section**
  - Renamed the model in Table 3 from *MCTS* to *MCTS-GNN* for consistency with Table 2.
  - Updated the *"Pauli's Exclusion Principle"* to *"Electrostatic Repulsion"* for better accuracy. Added the physics constraints explanations in Appendix Section D.5.
  - Included practical applications of PCGSR's solution in Appendix Section D.2.
  - Added the physical significance of the compressed copper dataset in Appendix Section D.7.

We hope these revisions improve the clarity and comprehension of our contributions. Once again, we thank the reviewers for their prompt responses and insightful suggestions. Your feedback has been invaluable in refining our work.

---

### Author Response · Authors · 2024-12-02
**Follow-Up**

We sincerely thank all the reviewers for their time and effort during the discussion period. We have carefully addressed all the concerns and questions raised in the reviews by providing detailed clarifications, additional analyses, new experimental results, and a thoroughly revised manuscript. These revisions, we believe, have significantly enhanced the quality and clarity of our work.

We would greatly appreciate any further feedback the reviewers may have regarding our latest responses and the updated manuscript. If there are any remaining questions or concerns, we are eager to address them promptly.

Once again, we extend our gratitude to all the reviewers for their insightful feedback and constructive suggestions throughout this process.

---

### Meta-Review · Area_Chair_Ynkc · 2024-12-20

**Metareview:**

This paper introduces "Graph Symbolic Regression" (SGN), which replaces traditional expression trees with a Symbolic Graph and employs a Graph Neural Network (GNN) to guide the Monte Carlo Tree Search (MCTS) algorithm. Experiments on benchmark datasets, synthetic data, and real-world applications demonstrate the effectiveness of the proposed method.

However, two reviewers have recommended rejection, and the authors' rebuttal does not fully address their concerns. First, the novelty of the approach is unclear, as it builds on well-established methods without offering a substantial new contribution. Second, the experiments are unconvincing, with insufficient datasets and scenarios to fully evaluate the robustness of the method.

In conclusion, I do not recommend accepting this paper in its current form.

**Additional Comments On Reviewer Discussion:**

Reviewers QFds, ER9J, and 4ixR rated this paper as 3: reject (raised to 5), 8: accept (kept the score), and 3: reject (kept the score), respectively.

The reviewers raised the following concerns:

- Lack of innovation (Reviewer 4ixR)

- Unclear motivation for the method (Reviewers QFds, 4ixR)

- Unclear explanation (Reviewer ER9J)

- Insufficient experiments (Reviewers QFds, 4ixR)

In the rebuttal, the authors addressed some concerns by providing additional experiments and clarifying certain aspects of the explanation. However, several key issues remain inadequately addressed. First, the paper lacks novelty, as the approach builds upon well-known methods without offering a significant new contribution. Second, the experiments are unconvincing, with insufficient datasets and scenarios to fully assess the method's robustness.
Therefore, I will not recommend accepting this paper in its current state.

---

### Decision · Program_Chairs · 2025-01-22

Reject